# New insights into the all-testis differentiation in zebrafish with compromised endogenous androgen and estrogen synthesis

Yonglin Ruan[1,2], Xuehui Li[1,2], Xinyi Wang[1,2], Gang Zhai[1,2]*, Qiyong Lou[1], Xia Jin[1], Jiangyan He[1], Jie Mei[3], Wuhan Xiao[1,2,4], Jianfang Gui[1,2,4,5], Zhan Yin[1,2,4,5]*

1 State key Laboratory of Freshwater Ecology and Biotechnology, Institute of Hydrobiology, Chinese Academy of Sciences, Wuhan, China, 2 College of Advanced Agricultural Sciences, University of Chinese Academy of Sciences, Beijing, China, 3 College of Fisheries, Huazhong Agricultural University, Wuhan, China, 4 The Innovative Academy of Seed Design, Chinese Academy of Sciences, Wuhan, China, 5 Hubei Hongshan Laboratory, Huazhong Agriculture University, Wuhan, China

* zhaigang@ihb.ac.cn (GZ); zyin@ihb.ac.cn (ZY)

## Abstract

The regulatory mechanism of gonadal sex differentiation, which is complex and regulated by multiple factors, remains poorly understood in teleosts. Recently, we have shown that compromised androgen and estrogen synthesis with increased progestin leads to all-male differentiation with proper testis development and spermatogenesis in *cytochrome P450 17a1* (*cyp17a1*)-/- zebrafish. In the present study, the phenotypes of female-biased sex ratio were positively correlated with higher *Fanconi anemia complementation group L* (*fancl*) expression in the gonads of *doublesex and mab-3 related transcription factor 1* (*dmrt1*)-/- and *cyp17a1*-/-;*dmrt1*-/- fish. The additional depletion of *fancl* in *cyp17a1*-/-;*dmrt1*-/- zebrafish reversed the gonadal sex differentiation from all-ovary to all-testis (in *cyp17a1*-/-; *dmrt1*-/-;*fancl*-/- fish). Luciferase assay revealed a synergistic inhibitory effect of Dmrt1 and androgen signaling on *fancl* transcription. Furthermore, an interaction between Fancl and the apoptotic factor Tumour protein p53 (Tp53) was found *in vitro*. The interaction between Fancl and Tp53 was observed via the WD repeat domain (WDR) and C-terminal domain (CTD) of Fancl and the DNA binding domain (DBD) of Tp53, leading to the K48-linked polyubiquitination degradation of Tp53 activated by the ubiquitin ligase, Fancl. Our results show that testis fate in *cyp17a1*-/- fish is determined by Dmrt1, which is thought to stabilize Tp53 by inhibiting *fancl* transcription during the critical stage of sexual fate determination in zebrafish.

## Author summary

Gonadal sex differentiation is known as the queen of problems in evolutionary biology, and the mechanisms that determine sexual fate vary widely among teleosts. Traditionally, Dmrt1 and androgen signaling have been essential for testis differentiation, and estrogen signaling has been essential for ovary differentiation. The all-testis phenotype observed in

**Data Availability Statement:** The RNA-seq raw data are available from the BioProject database (accession number: PRJNA999043). The other

relevant data are within the paper and its
Supporting Information files.

**Funding:** This work was supported by the National
Natural Science Foundation, China (32230108 to
ZY), National Key Research and Development
Program, China (2022YFD2401800 to GZ),
National Natural Science Foundation, China
(31972779 to GZ), Foundation of Hubei Hongshan
Laboratory (2021hszd021 to ZY and 2021hskf013
to GZ), Pilot Program A Project from the Chinese
Academy of Sciences (XDA24010206 to ZY),
Youth Innovation Promotion Association of CAS
(2020336 to GZ), and State Key Laboratory of
Freshwater Ecology and Biotechnology
(2016FBZ05 to ZY). The funders had no role in
study design, data collection and analysis, decision
to publish, or preparation of the manuscript.

**Competing interests:** The authors have declared
that no competing interests exist.

*cyp17a1-/-* zebrafish, has led to the conclusion that androgen signaling is dispensable for testis differentiation in zebrafish. By analyzing a series of mutant zebrafish lines, we show here that Dmrt1 sufficiently promotes all-testis differentiation in *cyp17a1*-deficient zebrafish albeit the compromised androgen and estrogen synthesis. In addition, we observed that zebrafish Dmrt1 and androgen signaling probably stabilize Tp53 by inhibiting the transcription of a ubiquitin ligase, Fancl. Our current study provides new insights into the interactive signals that regulate sexual fate determination in teleosts.

## Introduction

In vertebrates, the undifferentiated gonads rely on genetic and environmental sex determination (GSD and ESD) to determine their differentiation along male or female differentiation pathways [1,2]. In teleosts, the GSD are complex due to diversity within the species [3], and diverse master sex determination genes, including *dmrt1*, *dmrt1bY* (*DMY*), *anti-Müllerian hormone Y* (*amhy*), *gsdfY*, *etc* [4–9]. Additionally, some teleosts may ultimately tip the bipotential gonads towards the male or female fate in response to a continuum of genetic and environmental factors [10,11]. The domesticated experimental strains of zebrafish have lost their natural sex determinants, and lack a single strong genetic determinant [12].

Gonadal sex differentiation is further influenced by sex steroid hormones, especially androgens and estrogens, which are produced by the different steroidogenic lineages in the somatic cells of the testes or ovaries [13]. Genes encoding enzymes involved in sex steroid synthesis are differentially expressed during gonadal differentiation [3]. Cyp19a1a, an aromatase, converts testosterone to the estrogen 17β-estradiol (E2). Estrogen is known to be essential for ovary differentiation and maintenance, as indicated by the all-male phenotype observed in *cyp19a1a-/-*deficient zebrafish [14–16]. Additional depletion of *doublesex and mab-3 related transcription factor 1* (*dmrt1*), which drives male differentiation and maintains testis development, in *cyp19a1a-/-*deficient zebrafish resulted in partial ovary differentiation [17–19]. Cyp17a1, a cytochrome P450 enzyme with 17-alpha-hydroxylase and C17,20-lyase activities, is the key enzyme in the production of androgen and estrogen in animals [20]. Depletion of *cyp17a1* in zebrafish leads to all-testis differentiation, loss of male-typical secondary sex characteristics and mating behavior, due to their impaired androgen production [21–23]. Similarly phenotypes are also observed in *cyp17a1-/-* common carp and *scl* (*sex-character-less*, *P450c17*) mutant medaka, irrespective of the individuals' sex-determining genotypes (XY or XX) [24,25]. The increased progestin signaling seen in *cyp17a1-/-* fish was found to be responsible for proper testis development and spermatogenesis, which is dependent on the nuclear Progestin receptor (nPgr). It has also been shown that the increased endogenous progestin signaling does not have any effect on the sexual differentiation of the *cyp17a1-/-* zebrafish [23]. Theoretically, the double knockout of *cyp17a1* and *dmrt1* could further help to elucidate the mechanism underlying all-testis differentiation in *cyp17a1-/-* zebrafish with impaired endogenous androgen and estrogen signaling.

Zebrafish gonads initially form an ovary-like structure (called a "bipotential juvenile ovary"), which then develops into either into the ovaries in females or testes in males [26,27]. Gonadal differentiation follows oocyte apoptosis in the bipotential juvenile ovary during a critical window of time (20–30 days post-fertilization, dpf) that lasts for several days [27–29]. In zebrafish, the number of germ cells influences gonadal sex differentiation [30,31]. The complete loss of germ cells in *dnd* morphants leads to an all-male, sterile zebrafish phenotype [28,32,33]. It has been suggested that adult females can undergo sex reversal from female to

male when oocytes in the mature ovary are depleted, as observed in *nanos3*-null mutants or *ziwi*-CFP-NTR transgenic zebrafish at age of 5 months post-fertilization (mpf) [30]; Apparently, a sufficient number of germ cells is essential for the female differentiation in zebrafish [31,32]. Therefore, specific levels of oocyte-derived signals are thought to act on somatic cells in the gonads to promote and maintain ovary differentiation in zebrafish.

Fancl plays an important role in several biological processes, including DNA damage and apoptosis. In humans, *fancl* mutation leads to Fanconi Anemia (FA), which is a disease characterized by failure of bone marrow production, risk of developing cancer, hypogonadism, and impaired fertility [34]. Mutations in the FA pathway genes disrupt the repair of DNA damage caused by DNA inter-strand crosslinking [35]. Among FA genes, *fancc*, *fancg*, *fanca*, *fancd1* (*brac2*), and *fancd2* are known to cause hypogonadism, impaired gametogenesis, and infertility [36]. In zebrafish, a complete female-to-male sex reversal was observed for 12 of the 17 FA mutants, including *fancd1* and *fancj* homozygous knockout fish that were infertile [35,37]. *Fancl* transcripts begin to be observed in immature gonads at 17 and 23 dpf; their expression levels increase in developing germ cells at 26 dpf and persist to increase in developing oocytes and spermatocytes at 33 and 37 dpf [38]. The insufficient number of surviving oocytes caused by *fancl* depletion masculinizes the gonads in mutant zebrafish [37]. One factor known to mediate apoptosis in the developing gonad is Tp53 [39]. Specifically for male differentiation, Tp53-mediated apoptosis is required for the all-male phenotype in *brac2-/-* or *fancl-/-* zebrafish, as the all-male phenotype caused by increased apoptosis in the gonads could be rescued by *tp53* mutation [37,38]. However, *tp53* depletion can only partially rescue the sex reversal phenotypes exhibited by *fancd1*, *fancl*, *fancr* and *fancp* mutant zebrafish [35,37,38,40]. These results suggest a possible link between certain FA genes and Tp53, with elusive mechanisms determining gonadal differentiation.

In this study, all-female differentiation was observed in *cyp17a1-/-;dmrt1-/-* fish accompanied by significantly up-regulated *fancl* expression. Dmrt1 and androgen signaling probably stabilize Tp53 via inhibiting the transcription of *fancl*, which interacts with Tp53 and activates its K48-linked ubiquitination. Assuming that similar regulatory relationships exist in the germline and gonads, our results provide mechanistic insight into a novel regulatory function of the interactive germline signals and gonadal somatic signals in teleost gonadal sex determination.

## Results

### All *cyp17a1-/-;dmrt1-/-* fish developed as females

The double heterozygotes (*cyp17a1+/-;dmrt1+/-*) among the F1 progeny were bred to generate *cyp17a1-/-;dmrt1-/-* fish (S1A and S1B Fig). F2 progeny genotyped at 90 dpf were subjected to anatomical and histological analyses. The results demonstrated that 52.94% and 47.06% of control fish developed into females and males, respectively (Fig 1A, 1B, 1G, 1H and 1M). At 90 dpf, all gonads of *cyp17a1-/-* fish developed into testes (Fig 1C, 1I and 1M), whereas gonads of mostly *dmrt1-/-* fish developed into ovaries (78.26%, N = 23) (Fig 1D, 1E, 1J, 1K and 1M). Strikingly, all gonads of *cyp17a1-/-;dmrt1-/-* fish differentiated into ovaries (100%, N = 14) (Fig 1F, 1L and 1M). The isolation and staging analysis of ovarian follicles were then conducted. In contrast to the ovaries of control females and *dmrt1-/-* females that contained follicles at the early vitellogenic (EV), middle vitellogenic (MV), and full grown (FG) stages, the ovaries of *cyp17a1-/-;dmrt1-/-* fish only contained follicles at the primary growth (PG) and previtellogenic (PV) stages (Fig 1N). Both the serum of *cyp17a1-/-* fish and *cyp17a1-/-;dmrt1-/-* fish exhibited decreased concentration of estradiol compared to that of the control females (Fig 1O). Analysis of sex ratios in fish at 50 dpf showed that the *cyp17a1-/-* fish were

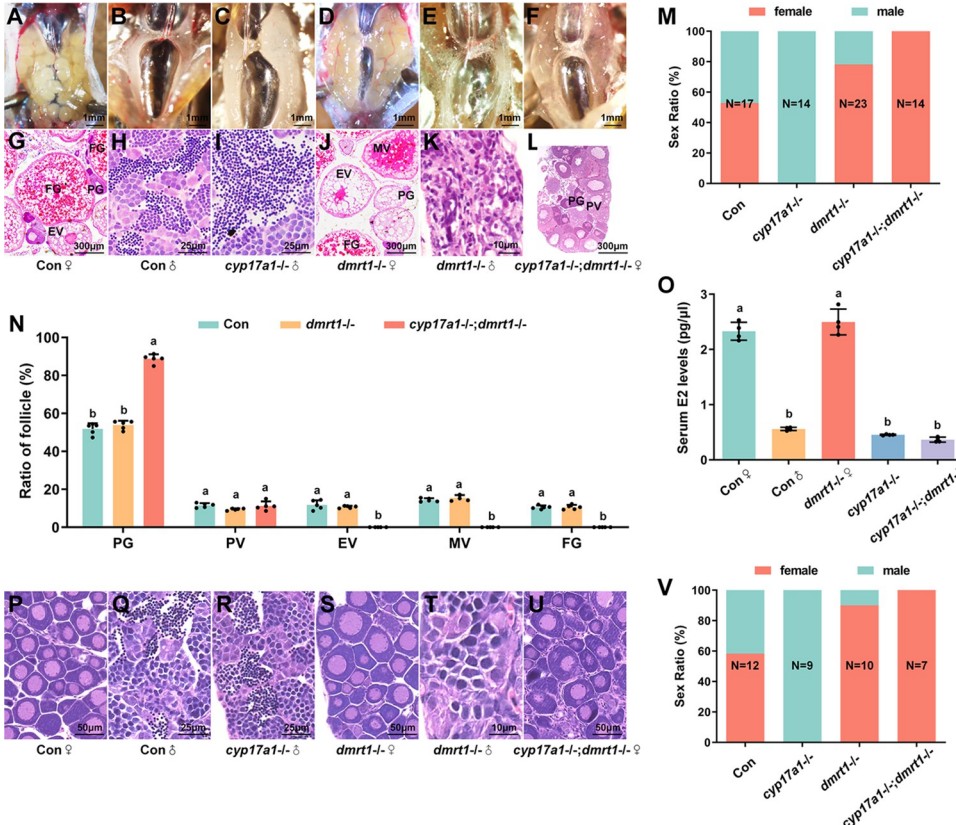

**Fig 1. Additional depletion of *dmrt1* restored the phenotype of ovary differentiation in *cyp17a1-/-* zebrafish.** (A–F) Anatomical examination of the gonads from the control fish, *cyp17a1-/-* fish, *dmrt1-/-* fish, and *cyp17a1-/-;dmrt1-/-* fish at 90 dpf. (G–L) Histological analysis of the gonads from the control fish, *cyp17a1-/-* fish, *dmrt1-/-* fish, and *cyp17a1-/-;dmrt1-/-* fish at 90 dpf. (A and G) Control fish ovary. (M) Sex ratios in fish of each genotype mentioned above at 90 dpf. (N) Ratios of PG, PV, EV, MV and FG follicles in fish of each genotype at 90 dpf. PG, primary growth. PV, previtellogenic. EV, early vitellogenic. MV, middle vitellogenic. FG, full grown. (O) Concentration of serum estradiol in control females, control males, *cyp17a1-/-* fish, *dmrt1-/-* females, and *cyp17a1-/-;dmrt1-/-* fish. E2, estradiol. (P–U) Histological analysis of the gonads from the control fish, *cyp17a1-/-* fish, *dmrt1-/-* fish, and *cyp17a1-/-;dmrt1-/-* fish at 50 dpf. (V) Sex ratios in fish of each genotype mentioned above at 50 dpf. Different letters in the bar charts represent significant differences.

all males, whereas all of the *cyp17a1-/-;dmrt1-/-* fish developed into females (Fig 1R, 1U and 1V). Compared with the control and *dmrt1-/-* female fish, PG follicles of the early folliculogenesis occurred normally in *cyp17a1-/-;dmrt1-/-* fish (Fig 1P, 1S and 1U). Based on these results, we postulated that a signal other than estrogen signaling induce ovary differentiation in *cyp17a1-/-;dmrt1-/-* fish.

## Increased expression of *fancl* in the gonad of *cyp17a1-/-;dmrt1-/-* fish

To identify the genes most likely regulate ovary differentiation of *cyp17a1-/-;dmrt1-/-* fish, which displayed an all-female differentiation, the candidate genes were selected based on the previous transcriptome analyses of the dissected gonads from presumptive female and male wildtype fish at 25 and 30 dpf [41]. Among these candidate genes, *fancl* was selected based on its early expression in gonads at 17 and 23 dpf [38], and its abundant expression in presumptive ovaries [41]. This aligns with our hypothesis that the gene(s) responsible for the all-female differentiation exhibited by *cyp17a1-/-;dmrt1-/-* fish should be specifically expressed in the

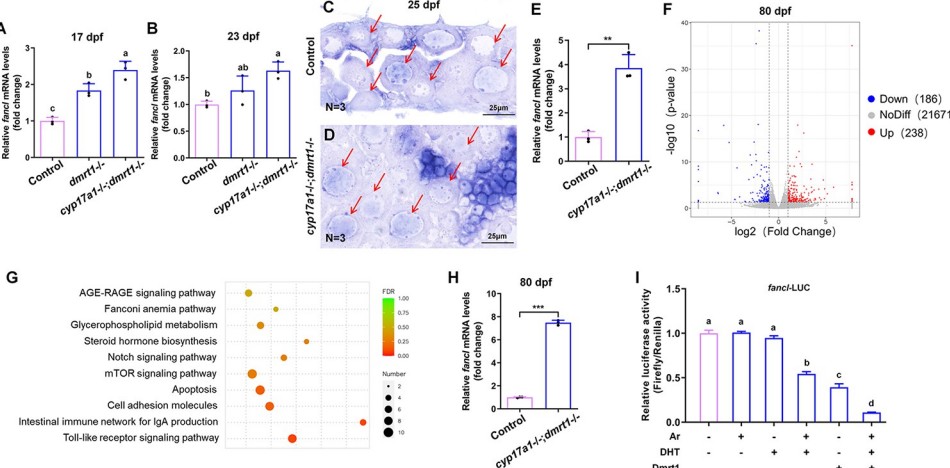

**Fig 2. The *cyp17a1-/-;dmrt1-/-* fish exhibited increased expression of *fancl*.** (A and B) Relative expression of *fancl* in control fish, *dmrt1-/-* fish and *cyp17a1-/-;dmrt1-/-* fish at 17 dpf and 23 dpf was tested with qPCR. For fish RNA sampling at 17 dpf and 23 dpf, every 5 body trunks of fish collected were mixed into one sample, and 3 samples were examined. (C and D) *In situ* hybridization was performed on cryosections of presumptive ovaries from control fish and *cyp17a1-/-;dmrt1-/-* fish at 25 dpf using the antisense probe of *fancl*. Arrows point to the immature oocytes. (E) Comparison of differentially expressed gene in the ovaries of control fish and *cyp17a1-/-;dmrt1-/-* fish at 80 dpf. Volcano plot shows genes that were differentially expressed in ovaries between control and *cyp17a1-/-;dmrt1-/-* fish. (F) Gene set enrichment analysis based on genes differentiated expressed in *cyp17a1-/-;dmrt1-/-* fish at 80 dpf. The length of the bar represents the false discovery rate (FDR). (G) Relative expression of *fancl* in the ovaries of control fish and *cyp17a1-/-;dmrt1-/-* fish at 80 dpf with qPCR. For fish at 80 dpf, every three dissected ovaries were mixed as one sample, and 3 samples were examined. (H) The effect of Dmrt1 and DHT/Ar in regulating the relative luciferase activity driven by *fancl* promoter. DHT, dihydrotestosterone. ***, p < 0.001. Different letters in the bar chart represent significant differences.

gonads during the critical period of zebrafish gonad differentiation and sex determination (17 to 33 dpf).

In support of this hypothesis, up-regulation of *fancl* was observed in the *cyp17a1-/-;dmrt1-/-* fish compared to the control females as verified by qPCR at 17 and 23 dpf (Fig 2A and 2B). Moreover, significantly up-regulated *fancl* expression was observed in *dmrt1-/-* fish at 17 dpf, while *fancl* up-regulation was not significant in *dmrt1-/-* fish at 23 dpf (Fig 2A and 2B). As *cyp17a1-/-;dmrt1-/-* fish all developed as females, we analyzed the expression level of *fancl* by *in situ* hybridization in presumptive ovaries from control fish and *cyp17a1-/-;dmrt1-/-* fish at 25 dpf. Compared to the moderate expression of *fancl* in the presumptive ovary of control fish, that of *cyp17a1-/-;dmrt1-/-* fish was significantly higher (Fig 2C–2E). *In situ* hybridization using the sense probe of *fancl* was also performed on cryosections of presumptive ovaries and as expected, no signals were detected (S2A and S2B Fig).

Comparative transcriptomic analyses were performed between ovaries from *cyp17a1-/-;dmrt1-/-* fish and *cyp17a1+/+;dmrt1+/+* female control siblings at 80 dpf. Compared to the ovaries from control fish, those from *cyp17a1-/-;dmrt1-/-* females exhibited significant expression level alterations of 424 genes (Fig 2F), with 238 genes being up-regulated and 186 genes being down regulated. The most enriched KEGG pathways in *cyp17a1-/-;dmrt1-/-* fish were significantly up-regulated as shown in Fig 2G. The top enriched pathways were related to intestinal immune network for IgA production, steroid hormone biosynthesis, Toll-like receptor signaling, Notch signaling and FA signaling (*fancl*). The up-regulated *fancl* expression in *cyp17a1-/-;dmrt1-/-* fish ovaries at 80 dpf was further verified by qPCR (Fig 2H). These differentially expressed genes may have important functions in ovary differentiation of *cyp17a1-/-;*

*dmrt1-/-* fish, although it still could not be excluded that their high expression was caused by their expression in primary oocytes or lower abundance in growing and mature oocytes.

To inspect whether Dmrt1 and androgen signaling could transcriptionally regulate *fancl* expression, a 2.5 kb region upstream of the zebrafish *fancl* transcription site was cloned into pGL3-basic vector. Both Dmrt1 and dihydrotestosterone (DHT)/Androgen receptor (Ar) inhibit the relative luciferase activity driven by *fancl* promoter, and their combinational treatment resulted in the largest inhibitory effect *in vitro* (Fig 2I).

## Increased *fancl* expression sustained ovary differentiation in *cyp17a1-/-*; *dmrt1-/-* females

We then set out to verify whether increased *fancl* expression sustained female differentiation in *dmrt1-/-* or *cyp17a1-/-;dmrt1-/-* females. The *cyp17a1-/-;dmrt1-/-;fancl-/-* fish was generated by mating triple heterozygotes (*cyp17a1+/-;dmrt1+/-;fancl+/-*) (S3A–S3D Fig). Anatomical analyses of gonad differentiation in samples obtained from *dmrt1-/-* fish, *cyp17a1-/-;fancl-/-* fish, *cyp17a1-/-;dmrt1-/-;fancl-/-* fish, and control siblings were conducted. Again, 52.63% and 47.36% of the control fish developed into females and males, respectively (Fig 3A, 3B, 3G, 3H and 3M), whereas *dmrt1-/-* fish mostly developed into females (78.95%, N = 19) (Fig 3C, 3D, 3I, 3J and 3M). All-testis differentiation was observed in *cyp17a1-/-;fancl-/-* fish (100.00%, N = 21) (Fig 3E, 3K and 3M). Unlike *cyp17a1-/-;dmrt1-/-* fish which developed as ovaries, the *cyp17a1-/-;dmrt1-/-;fancl-/-* fish developed as testis with histological apparent abnormalities including fibroblast-like somatic cells and diffuse vacuolation, similar to those observed in *cyp17a1-/-;dmrt1-/-* fish (100.00%, N = 10) (Fig 3F, 3L and 3M), which resembles the observations in the testes of *dmrt1-/-* fish (Fig 3J) [19, 42]. Accordingly, dissected testis of *dmrt1-/-* fish and *cyp17a1-/-;dmrt1-/-;fancl-/-* fish were hypoplastic compared to controls (Fig 3N–3P).

We observed that the testes of *dmrt1-/-;fancl-/-* fish were also hypoplastic and lack germ cells similar to *dmrt1-/-* fish, compared to the control fish (Fig 3Q–3C1). The results highlighting the antagonistic role of Fancl and Dmrt1 in determining gonadal sex, not only suggest that Fancl is required for ovary differentiation in *dmrt1-/-;fancl-/-* fish, but also imply the existence of other Dmrt1 targets required for testis development, as the testis is impaired in *dmrt1-/-; fancl-/-* fish.

## Zebrafish Fancl interacts with Tp53 *in vitro*

Fancl plays an important role in the survival of developing oocytes during meiosis, and Tp53-mediated germ cell apoptosis induces sex reversal in *fancl* mutant zebrafish [38]. Given the higher expression of *fancl* in the gonadal tissue of *cyp17a1-/-;dmrt1-/-* fish, which is an all-ovary differentiation context, we postulated that Fancl may play a role in ovary differentiation by interacting with the Tp53 signaling cascade. To test this hypothesis, a co-immunoprecipitation assay was performed in HEK293T cells. Myc-tagged Fancl and Flag-tagged Tp53 plasmids were transfected into HEK293T cells. The interaction between exogenously Flag-tagged Tp53 and Myc-tagged Fancl was observed by a reciprocal Co-immunoprecipitation (Co-IP) experiment (Fig 4A). Domain mapping of the interaction between Fancl and Tp53 indicated that the DNA-binding domain (DBD) of Tp53 is required for their interaction (Fig 4B and 4C). The multiple domains, WD-repeat domain (WDR) and C-terminal domain (CTD), of Fancl are required for their interaction, rather than single functional domain (Figs 4D, 4E, S4A and S4B).

## Zebrafish Fancl activated K48-linked ubiquitination of Tp53 *in vitro*

Fancl is a member of the Fanconi Anemia core complex with a plant homeodomain (PHD) that mono-ubiquitinates Fancd2 and Fanci [34]. The experimental results from western

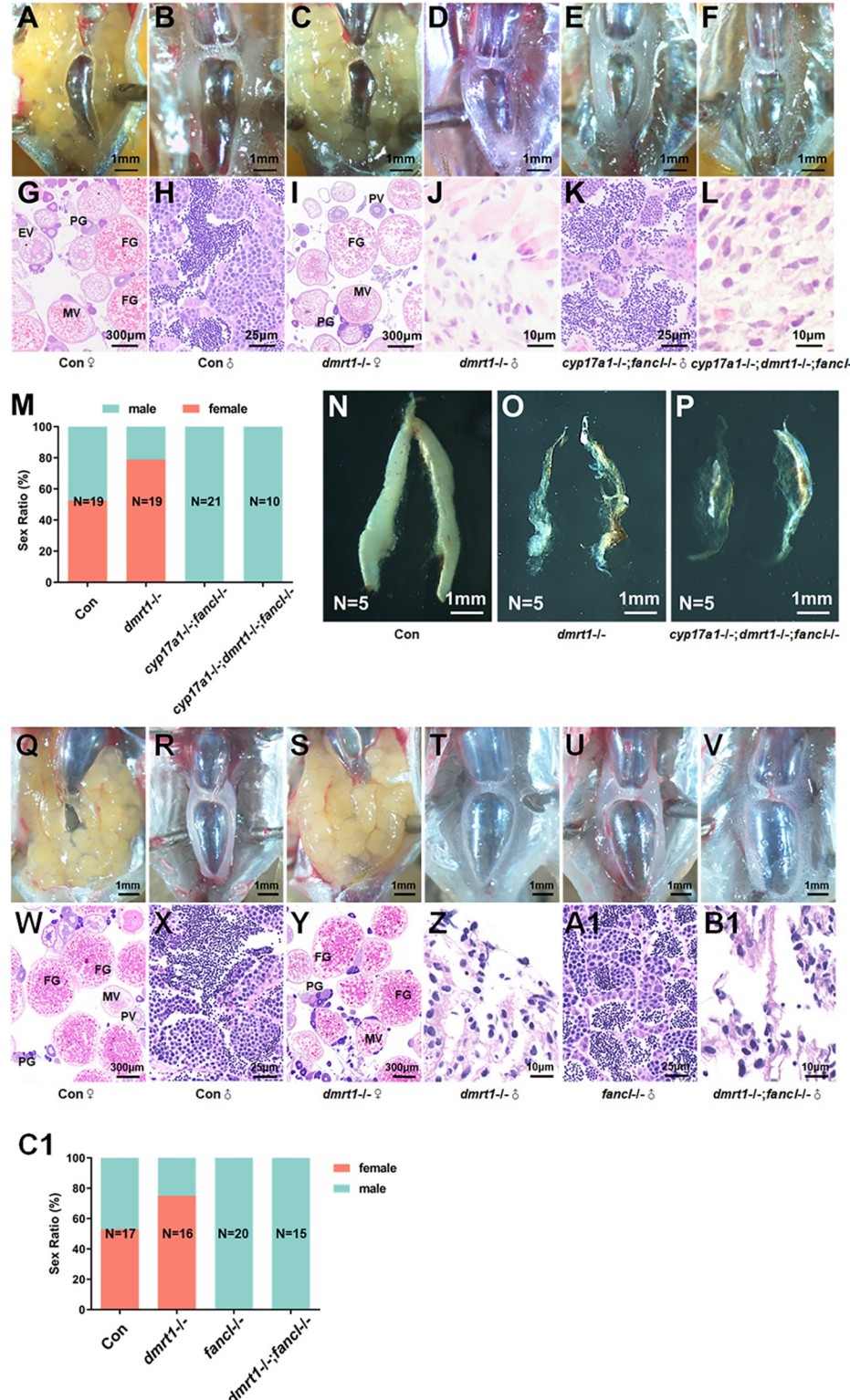

**Fig 3. Additional depletion of *fancl* blocked female-biased sex ratio of *cyp17a1-/-;dmrt1-/-* fish and *dmrt1-/-* fish.**
(A–F) Anatomical examination of the gonads from the control fish, *dmrt1-/-* fish, *cyp17a1-/-;fancl-/-* fish, and *cyp17a1-/-;dmrt1-/-;fancl-/-* fish. (G–L) Histological analysis of the gonads from the control fish, *dmrt1-/-* fish, *cyp17a1-/-;fancl-/-* fish and *cyp17a1-/-;dmrt1-/-;fancl-/-* fish. (M) Sex ratios in fish of each genotype mentioned above at 90 dpf. (N-P) The visualization of dissected testis of control fish, *dmrt1-/-* fish and *cyp17a1-/-;dmrt1-/-;fancl-/-* fish.

(Q–V) Anatomical examination of the gonads from the control fish, *dmrt1-/-* fish, *fancl-/-* fish, and *dmrt1-/-; fancl-/-* fish. (W–B1) Histological analysis of the gonads from the control fish, *dmrt1-/-* fish, *fancl-/-* fish, and *dmrt1-/-; fancl-/-* fish. (C1) Sex ratios in fish of each genotype mentioned above at 90 dpf. PG, primary growth. PV, previtellogenic. EV, early vitellogenic. MV, middle vitellogenic. FG, full growth.

blotting analysis of HEK293T cells transfected with Myc-tagged Fancl and Flag-tagged Tp53 demonstrated that transfection of Myc-tagged Fancl decreased the levels of Flag-tagged Tp53 in a dose-dependent manner (Fig 5A and 5B). When MG-132, a proteasome inhibitor, was present, Fancl-mediated Tp53 destabilization was effectively blocked (Fig 5C and 5D). We further demonstrated that Fancl increased K48-linked ubiquitination of Tp53 (Fig 5E). These

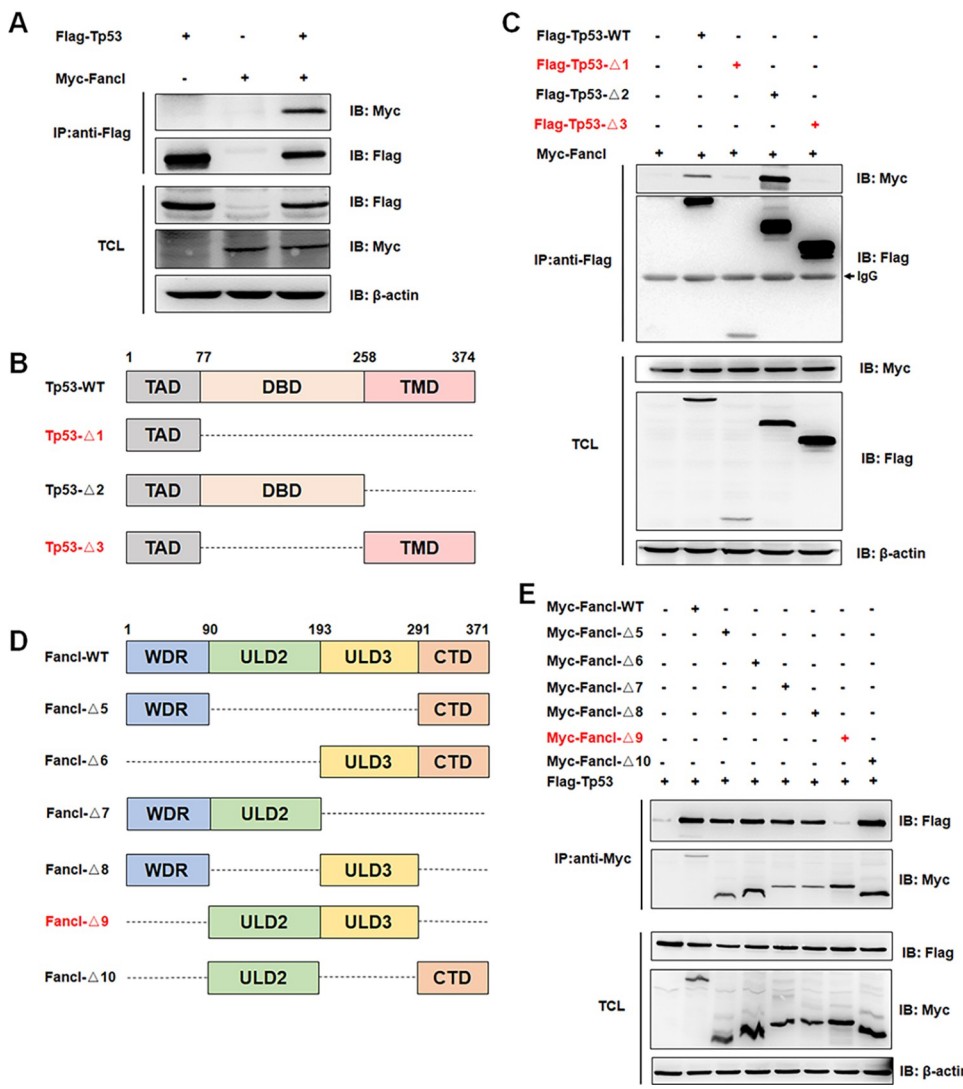

**Fig 4. Fancl interacts with Tp53 in HEK293T cells.** (A) The interaction of Fancl with Tp53 in HEK293T cells as revealed by the Co-IP assay. Myc-tagged Fancl and Flag-tagged Tp53 were transfected into HEK293T cells, then anti-Flag antibody-conjugated agarose beads were used for immune-precipitation. (B and C) Domain mapping revealed that the DBD domain of Tp53 is required for their interaction. TAD, transactivation domain. DBD, DNA binding domain. TMD, tetramerization domain. (D and E) Domain mapping revealed that the multiple domains, WDR and CTD, of Fancl are required for their interaction. WDR, WD-repeat domain. ULD2, UBC-like domain 2. ULD3, UBC-like domain 3. CTD, C-terminal domain. IP, immunoprecipitation. IB, immunoblotting. TCL, total cell lysate.

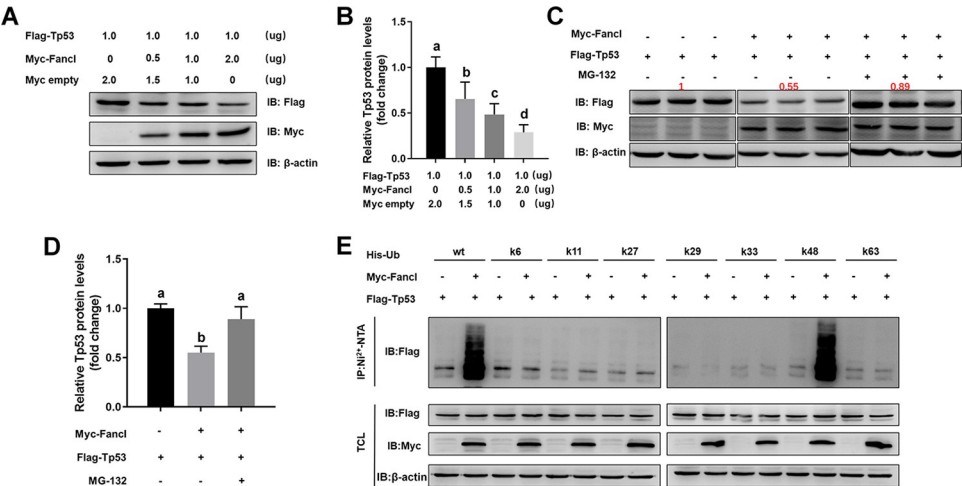

**Fig 5. Fancl promoted K48-linked ubiquitination of Tp53 in HEK293T cells.** (A) Transfection of Myc-tagged Fancl decreased the levels of Flag-tagged Tp53 in a dose-dependent manner. (B) Quantification of the western blot bands of the target protein, Tp53. (C) The proteasome inhibitor, MG-132, blocked the Fancl-mediated Tp53 destabilization. (D) Quantification of the western blot bands of the target protein, Tp53. (E) Fancl promotes K48-linked ubiquitination, rather than K6-, K11-, K27-, K29-, K33-, K63-linked ubiquitination of Tp53. IP, immunoprecipitation. IB, immunoblotting. TCL, total cell lysate. Different letters in the bar charts represent significant differences.

results suggest that Fancl promotes K48-linked ubiquitination, rather than K6-, K11-, K27-, K29-, K33-, and K63-linked ubiquitination of Tp53.

## Arrested follicular development in *cyp17a1-/-;dmrt1-/-* females was rescued by 17β-estradiol

The results described above demonstrated that all the gonads of *cyp17a1-/-;dmrt1-/-* fish differentiated into ovaries. To identity whether supplementation with estrogen could improve the arrested follicular development of *cyp17a1-/-;dmrt1-/-* females, treatment with 17β-estradiol was performed in *cyp17a1-/-;dmrt1-/-* fish from 80 to 110 dpf. Control females at 110 dpf exhibited normal ovaries containing follicles at the vitellogenic (EV+) stage (Fig 6A and 6D), and follicles of the *cyp17a1-/-;dmrt1-/-* fish were arrested at PG and PV stages (Fig 6B and 6E). However, EV+ follicles were observed in *cyp17a1-/-;dmrt1-/-* fish after 0.1 μg/L 17β-estradiol administration (Fig 6C, 6F and 6G). These results indicate that 17β-estradiol treatment effectively rescued the arrested folliculogenesis prior the EV+ stage of *cyp17a1-/-;dmrt1-/-* fish.

## Discussion

The key functions of estradiol in zebrafish gonadal sex determination have been extensively documented [14,15,21]. In *cyp17a1-/-* zebrafish and common carp, it would be interesting to explore the mechanisms underlying the all-testis differentiation and successful spermatogenesis [21,25]. Augmentation of progestin signaling has been proposed to be responsible for the proper testis organization and spermatogenesis in *cyp17a1-/-;ar-/-* zebrafish. However, additional depletion of *npgr* in *cyp17a1-/-;ar-/-* fish leads to the phenotypes of all-testis differentiation with impaired spermatogenesis in *cyp17a1-/-;ar-/-;npgr-/-* zebrafish [23]. This result suggests progestin is important for normal organization of the testis and spermatogenesis, but not for determination of testis fate [23].

The ovary fate of *cyp19a1a*-deficient zebrafish can be restored when *dmrt1* is additionally depleted (in *cyp19a1a-/-;dmrt1-/-* fish), suggesting an antagonistic function of Cyp19a1a and

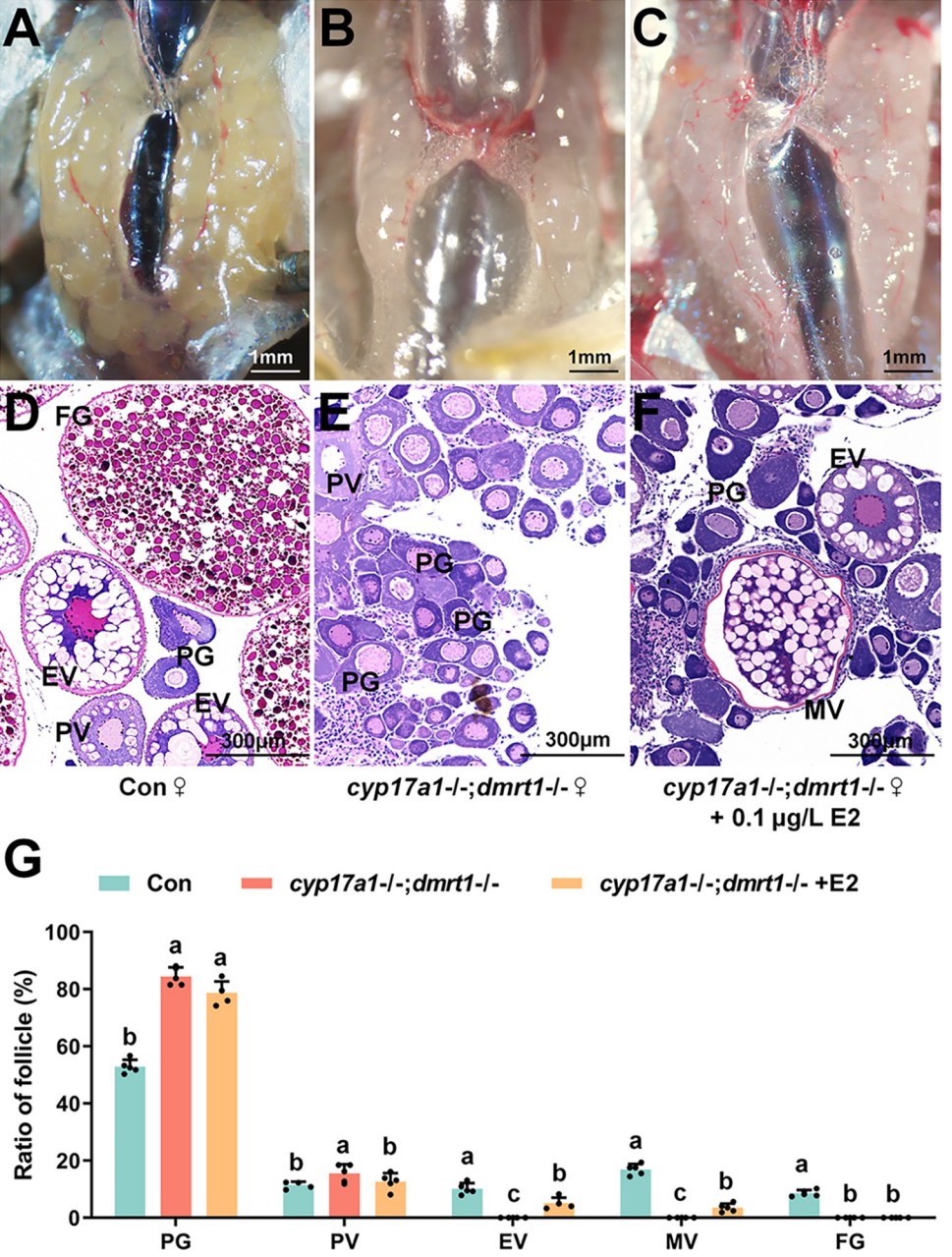

**Fig 6. Administration of 17β-estradiol rescued the arrested folliculogenesis of *cyp17a1*-/-;*dmrt1*-/- fish.** (A–C) Histological analysis of the ovaries from the control fish, *cyp17a1*-/-;*dmrt1*-/- fish, and *cyp17a1*-/-;*dmrt1*-/- fish administrated 17β-estradiol from 80 to 110 dpf. (D–F) Anatomical examination of the ovaries from the control fish, *cyp17a1*-/-;*dmrt1*-/- fish and *cyp17a1*-/-;*dmrt1*-/- fish administrated 17β-estradiol from 80 to 110 dpf. (G) Ratios of PG, PV, EV, MV and FG follicles in fish of each genotype at 110 dpf. PG, primary growth. PV, previtellogenic. EV, early vitellogenic. MV, middle vitellogenic. FG, full grown. Different letters in the bar chart represent significant differences.

Dmrt1 in determining sexual fate in zebrafish [17,18]. Similarly, ovary differentiation in our *cyp17a1*-/-;*dmrt1*-/- zebrafish at 90 dpf indicates that Dmrt1 determines testis fate despite the testosterone- and estradiol-deficiency of *cyp17a1*-/- zebrafish. Ratio of the ovary maintenance in *cyp19a1a*-/-;*dmrt1*-/- zebrafish progressively decreases after 60 dpf, with female ratios of

83%, 43% and 13% at 60, 75 and 100 dpf, respectively [17]. In contrast to this, the ratio of the ovaries detected among our *cyp17a1-/-;dmrt1-/-* fish at 90 dpf was 100% (N = 14). The previously published profiles allowed us to compare the endogenous steroids concentration between *cyp17a1-/-* fish and *cyp19a1a-/-* fish. In *cyp17a1-/-* and *cyp19a1a-/-* fish, respectively, testosterone was found to be impaired and elevated [14, 21, 23]. Compared to *cyp19a1a-/-;dmrt1-/-* fish, it is reasonable to assume that the higher frequency of ovary differentiation displayed by *cyp17a1a-/-;dmrt1-/-* fish could be attributed to its androgen deficiency. In other words, the synergistic effects of simultaneous depletion of *dmrt1* and *cyp17a1* may promote ovary differentiation. This is also consistent with the ovarian biased differentiation phenotype reported in *ar-/-* zebrafish, suggesting that androgen signaling may indeed antagonize ovary differentiation in zebrafish [43–45].

In zebrafish, *fancl* was up-regulated in the presumptive ovaries as compared with the presumptive testes [41]. In our analyses, the increased expression of *fancl* in presumptive ovaries of *cyp17a-/-;dmrt1-/-* fish at 17, 23 and 25 dpf compared to that in control females was positively correlated with their female biased sex ratio. Indeed, we also observed a moderate up-regulation of *fancl* in *dmrt1-/-* fish at 17–23 dpf; however, it is not as significant as in *cyp17a-/-;dmrt1-/-* fish. This is unlikely to be driven by differences in the stages of oocytes present in the ovaries in our different mutant contexts, as the *dmrt1-/-* female zebrafish is fertile and develops normally [42]. The observed synergy between Dmrt1 and androgen signaling in inhibiting *fancl* transcription in luciferase reporter assays suggests that elevated *fancl* in *cyp17a1-/-;dmrt1-/-* fish may result from loss of these synergistic repressors.

Disruption of *fancl* is known to cause masculinized gonads and testis differentiation due to increased germ cell apoptosis compromises oocyte survival, which could be rescued by *tp53*-depletion [38]. Fancl is known as a ubiquitin ligase [34]. Our *in vitro* results indicate that zebrafish Fancl interacts with Tp53 to promote its degradation through K48-linked polyubiquitination. Assuming that similar regulatory relationships exist in the germline and gonads *in vivo*, these results provide new insights into the regulatory network involved in Fancl functions for ovary differentiation, which may protect germ cells from apoptosis induced by Tp53 signaling in zebrafish. Notably, increased apoptosis and decreased Vasa-positive germ cells were observed with *dmrt1* deficiency previously [42]. Additional depletion of *fancl* resulted in all-testes differentiation in *cyp17a1-/-;dmrt1-/-* fish, due to impaired ovary differentiation. Therefore, it could be concluded that the Fancl-mediated germ cell survival is determinant in gonadal differentiation in *cyp17a1-/-;dmrt1-/-* zebrafish.

The severely hypoplastic testes observed in *dmrt1-/-;fancl-/-* fish and *cyp17a1-/-;dmrt1-/-;fancl-/-* fish are consistent with our previous view that Dmrt1 is required for the maintenance of male germ cells [42]. This could be interpreted that additional *fancl* depletion promotes testis differentiation in *dmrt1-/-* fish and *cyp17a1-/-;dmrt1-/-* fish, but their male germ cell development is dysregulated due to Dmrt1 deficiency. Similar observations have been reported with zebrafish whole testis differentiation upon depletion of *RNA-binding protein of multiple splice forms 2* (*rbmps2*), a critical germline-expressed factor for female sex differentiation. The severe hypoplastic testes were observed in *dmrt1-/-;rbmps2a-/-;rbmps2b-/-* zebrafish [18]. Significantly, depletion of genes related to germ cell development and survival in fish results in the same phenotype of the lost germ cells as they attempt to embark on the male fate but suffer from the lack of Dmrt1 [18].

Loss of *tp53* can restore ovarian development in *fancd1*(*brca*), *fancl*, *fancp* and *fancr* mutant fish [35]. In contrast, introducing *tp53* mutation did not restore ovary differentiation in *dazl*, *figla*, *rbm46*, *vasa* and *rbpms2* mutants, which are key factors involved in germ cell survival, meiosis and differentiation [46–50]. We also observed that depletion of *tp53* (IHB136, China Zebrafish Resource Center) did not affect sexual differentiation in zebrafish and could not

restore ovary differentiation in *cyp17a1-/-* zebrafish. This suggests that germ cell loss is not exclusively mediated by apoptosis via Tp53. Further studies, such as generating and analyzing *dmrt1-/-;fancl-/-;tp53-/-* zebrafish, are needed to investigate the mechanisms of Dmrt1 on Fancl/Tp53 signaling on the link between sex fate determination and germ cell survival.

Increased progestin signaling maintains proper testis organization and spermatogenesis in *cyp17a1-/-;ar-/-* zebrafish, suggesting a dispensable role for androgen in testis organization and spermatogenesis under certain circumstances [23]. In contrast, only PG and PV stage follicles were observed in *cyp17a1-/-;dmrt1-/-* zebrafish at 90 dpf. Together with a similar follicle status observed in *cyp19a1a-/-;dmrt1-/-* zebrafish, the arrested follicle development from PV to EV transition with impaired vitellogenesis could be attributed to impaired estrogen synthesis when either *cyp19a1a* or *cyp17a1* was depleted in these mutants [17, 18]. The vitellogenic (EV+) follicles were observed in *cyp17a1-/-;dmrt1-/-* fish after 17β-estradiol administration from 80 to 110 dpf, supporting the essential function of 17β-estradiol in regulating folliculogenesis via vitellogenesis. Ovarian follicles with failed yolk accumulation were also observed in *cyp19a1a-/-* female medaka [51]. Taken together, the endogenous estrogens synthesized in zebrafish and medaka are dispensable for ovarian differentiation, but indispensable for ovarian development and oocyte maturation. On the other hand, although *cyp17a1* deficiency leads to elevated progestin levels (in *cyp17a1-/-* fish) [23], it is not sufficient to provide an adequate gonadal steroid hormone environment for folliculogenesis in *cyp17a1-/-;dmrt1-/-* zebrafish.

Determination of the fate of gonadal supporting cells in mammals plays a critical role in gonadal sex determination, as *Sry* acts spatiotemporally to switch supporting cells from the female to the male pathway [52]. Both in XX and XY mice, depletion for *CYP17A1* exclusively causes phenotypically female appearance (external genital phenotype), abnormal inner genitalia development and infertile phenotypes [53]. Compared to the phenotype observed in *CYP17A1*-null mice, all-testis differentiation and proper spermatogenesis were observed in *cyp17a1-/-* zebrafish and common carp [21, 25]. These results demonstrate the evolutionary plasticity of sex determination and gonadal development in vertebrates. Of course, further studies of our proposed regulatory mechanism are needed in other fish species or mammals, which, we believe, would undoubtedly broaden the knowledge underlying sex determination in teleosts.

## Materials and methods

### Ethics statement

All fish experiments were conducted in accordance with the Guiding Principles for the Care and Use of Laboratory Animals, and were approved by the Institute of Hydrobiology, Chinese Academy of Sciences (Approval ID: IHB 2013724).

### Zebrafish maintenance

Zebrafish (*Danio rerio*) were maintained as previously described [54]. Briefly, the fish were kept in a circulated water system and maintained under standard laboratory conditions at 28.5˚C with a light/dark cycle of 14/10 hours; the Fish were fed twice daily with freshly hatched brine shrimp.

### The knockout lines

The loss-of-function alleles of *cyp17a1* and *dmrt1* in zebrafish (mutated with 7 and 14 bp deletion in the first exon) generated by our group as previously described was used in this study [21, 42]. The *cyp17a1* heterozygote was bred with a *dmrt1* heterozygote of the opposite sex to

**Table 1. Primers used in this study.**

| Gene | Primer direction and sequence (5'-3') | Product size (bp) | Reference |
|---|---|---|---|
| qPCR | | | |
| *fancl* | F: GAACCCTGACTGCACTGTCCTAC | 232 | [38] |
| | R: GCTTTGGCGACTGGTTGGCAGAC | | |
| *β-actin* | F: ACTCAGGATGCGGAAACTGG | 118 | [55] |
| | R: AGGGCAAAGTGGTAAACGCT | | |
| Genotyping | | | |
| *cyp17a1* | F: GCAGTGCTGTTCAGAAGAGCT | 559 | [22] |
| | R: GGCAGTTCATTCTGCTCTGA | | |
| *dmrt1* | F: CGTTATCAAACCTCAGACCCTA | 549 | [42] |
| | R: TAGCCAAAGCAGTCAACAAT | | |
| *cyp17a1* | F: GACAGTCCTCCGCACATCTTC | 250 | This study |
| | R: ACCATATGCAGATGGGCC | | |
| *fancl* | F: CCAGCAGATCATCCACCATCC | 237 | This study |
| | R: GAGCTGCCTCTCACACGCAGG | | |
| Guide RNA sequences | | | |
| *cyp17a1* | GGATCTCCTTCGCATGATGG | | This study |
| *fancl* | GGATCTCCTTCGCATGATGG | | This study |
| Promoter amplification | | | |
| *fancl* | F: TTTACTAGGTATACTTGAAAC | 2500 | This study |
| | R: CCTAGCAAAGCGAAAGTAACTT | | |

F, Forward. R, Reverse. bp, base pair.

generate *cyp17a1/dmrt1* double heterozygous fish, which were then inbred to generate an off-spring population containing *cyp17a1-/-;dmrt1-/-* fish. The introduction of the *fancl* knockout into *cyp17a1+/-;dmrt1+/-* fish could not be achieved by breeding, as *fancl* and *cyp17a1* are both located on chromosome 13. To obtain triple heterozygotes (*cyp17a1+/-;dmrt1+/-;fancl+/-*), CRISPR/Cas9-mediated *cyp17a1* and *fancl* knockouts were performed in F2 embryos derived from mating between *dmrt1+/-* females and *dmrt1+/-* males. The triple heterozygotes were then inbred to generate triple homozygotes (*cyp17a1-/-;dmrt1-/-;fancl-/-*). The females observed in *cyp17a1+/+;dmrt1+/+*, *cyp17a1+/+;dmrt1-/-*, and *cyp17a1-/-;dmrt1-/-* fish were used for the anatomical analysis and histological analysis. The *cyp17a1+/+;dmrt1+/+* and *cyp17a1-/-;dmrt1-/-* fish at 17, 23, 25 and 80 dpf were used for gene expression analysis. The *dmrt1* heterozygous males and females were inbred to generate an offspring population containing *dmrt1-/-* fish [42]. To obtain the triple heterozygotes (*cyp17a1+/-;dmrt1+/-;fancl+/-*), the CRISPR/Cas9-mediated deletions of *cyp17a1* and *fancl* were performed in F2 embryos derived from the inbred *dmrt1-/-* fish for the generation of the triple heterozygotes (mutated *cyp17a1* with a 31 bp deletion in the first exon and *fancl* with a 37 bp deletion in the ninth exon). The triple heterozygotes were then inbred to generate the triple homozygotes (*cyp17a1-/-;dmrt1-/-;fancl-/-*). The guide RNA sequences for the knockout lines and the primers used for genotyping are listed in Table 1.

## Histological analysis

Hematoxylin and eosin staining was performed as previously described [21]. Dissected gonads were fixed in 4% paraformaldehyde in phosphate-buffered saline (PBS) at room temperature. Fixed samples were dehydrated, infiltrated, and embedded in paraffin for sectioning. Sections

(7 μm-thick) were stained with hematoxylin and eosin and visualized under a Nikon Eclipse Ni-U microscope (Nikon, Tokyo, Japan). The oocytes at 50 and 90 dpf are shown with with a scale bar of 50 and 300 μm, respectively. Normal testes are shown with a scale bar of 25 μm, and the hypoplastic testes are shown with a scale bar of 10 μm.

### *In situ* hybridization

The *in situ* hybridization on cryosections of presumptive ovaries were performed as previous described [56]. Sense and anti-sense digoxigenin-labeled cRNAs of *fancl* were synthesized and used in this study. The cDNA fragment of 786 nt containing the PHD domain of *fancl* was used to synthesize probe as previously described [38]. The *in situ* hybridization were photographed using a Nikon Eclipse Ni-U microscope (Nikon, Tokyo, Japan). Staining intensity in germ cells were quantified by analyzing the gray values using Image J software (National Institutes of Health, Bethesda, MD, USA). There were three independent replicates for the fish of each genotype.

### Isolation and staging of ovarian follicles

The staging system adopted for ovarian follicles was conducted based on the original definition of Selman *et al.* [57] as modified by the researchers [17, 58–61]. Ovaries, which were dissected out from females of each genotype after anesthetization, were placed in a 60 mm culture dish containing 60% L-15 medium (Gibco, Carlsbad, CA, United States). The follicles of different stages were manually isolated and divided into five stages according to their size and vitellogenic stage: primary growth stage (~0.1 mm), previtellogenic stage (cortical alveolus, ~0.3 mm), early vitellogenic stage (~0.4 mm), midvitellogenic stage (~0.5 mm), and full grown but immature stage (~0.65 mm).

### Serum estradiol measurement

The concentrations of serum estradiol in fish at 3 mpf were measured using commercial ELISA kit (582251, Cayman Chemical Company, Ann Arbor, MI) as previously described [21]. There were four independent replicates for the fish of each genotype.

### RNA extraction and quantitative real-time PCR (qPCR)

Total RNA was extracted from zebrafish using TRIzol reagent (15596–026; Invitrogen, Carlsbad, CA, United States) following a previously described standard protocol [21]. For zebrafish at 17 and 23 dpf, truncated zebrafish bodies, which contained the gonads, were used for RNA extraction. We synthesized cDNA using One-Step gDNA Removal and cDNA Synthesis SuperMix (AE311-02; Transgen Biotech, Beijing, China). qPCR was performed using SYBR Green Real-Time PCR Mix (AQ131-01; Transgen Biotech) and a real-time PCR system (Bio-Rad, Hercules, CA, United States). The housekeeping gene *β-actin* was used as endogenous control, and the expression level of *fancl* was calculated as the fold change relative to *β-actin* [62]. The primers used for qPCR are listed in Table 1. The *cyp17a1+/+;dmrt1+/+* female fish of the *cyp17a1-/-;dmrt1-/-* siblings served as control. The comparison was performed between the fish of different genotypes.

### Transcriptome analysis

At the indicated time points, total RNA from the dissected ovaries was extracted using TRIzol reagent. RNA-seq reads were generated using the Illumina NovaSeq 6000 system. High-quality mRNA reads were mapped to the *Danio rerio* genome (GRCz11) using HISAT2 (version 2.2.4,

http://daehwankimlab.github.io/hisat2/). Differential expression analysis was performed using the DESeq2 package (v1.30.1) with a fold change of two and a p-value cutoff of 0.05.

## Plasmid constructions

Zebrafish Fancl, Tp53, Dmrt1 and Ar were cloned into the vectors of pCMV-myc modified pCMV-flag, and pcDNA3.1(+), respectively. For *fancl* luciferase construction, a 2.5 kb region upstream of the transcription initiation site of zebrafish fancl was cloned into the pGL3-basic plasmid. The primers used for *fancl* promoter amplification are listed in Table 1.

## Cell culture and transfection

Human embryonic kidney (HEK) 293T cells (originally obtained from American Type Culture Collection, Manassas, VA, United States) were grown at 37˚C in a humidified incubator containing 5% $CO_2$ in high glucose Dulbecco's Modified Eagle's Medium (DMEM) (06-1055-57-1A; BI, Israel) supplemented with 10% fetal bovine serum (FBS). Plates at 60% confluency were transfected with X-tremeGene HP (6366236001; Roche, Basel, Switzerland) according to the manufacturer's instructions. After 12 h post transfection, cells were recovered to the culture medium containing dimethyl sulfoxide (DMSO) or DHT (10 nmol/L) (D413176, Aladdin, Shanghai, China), and harvested at 24 h post transfection.

## Luciferase reporter assay

Luciferase activity was measured using the dual-luciferase reporter assay system following the manufacturer's instructions (E1910, Promega, Madison, WI, United States). Data were normalized to Renilla luciferase. The relative luciferase activity in transfected cells was detected using a Sirius Luminometer from Berthold Detection Systems. Data were obtained from three independent experiments.

## Western blotting

Total protein content from HEK 293T cells was extracted with RIPA buffer containing 50 mM Tris (pH 7.4), 1% NP-40, 0.25% sodium deoxycholate, 1 mM EDTA (pH 8), 150 mM NaCl, 1 mM NaF, 1 mM PMSF, 1 mM Na3VO4, and a 1:50 dilution of the protease inhibitor mixture (P1045; Beyotime, Shanghai, China). Then, the proteins were separated by sodium dodecyl sulfate-polyacrylamide gel electrophoresis and transferred on a polyvinylidene fluoride (PVDF) membrane. Mouse anti-Myc (1:1000, Santa Cruz, Dallas, TX, United States), mouse anti-Flag (1:1000, Sigma-Aldrich, St. Louis, MO, United States), and rabbit anti-β-Actin (1:1000, Abclonal, Wuhan, China) were used as the primary antibodies. Horseradish peroxidase (HRP) conjugated anti-mouse (SA00001-1; Proteintech, Wuhan, China) and rabbit secondary (AS014; Abclonal Wuhan, China) antibodies were used at a 1:5000 dilution. The membranes were stained with Immobilon Western Chemiluminescent HRP substrate (WBKLS0500; Millipore, Billerica, MA, United States) and detected by using an ImageQuant LAS 4000 system (GE Healthcare, Fairfield, MA, United States).

## Co-IP analysis

For Co-IP analysis, HEK 293T cells grown to 60% confluency were transfected with a total of 10 μg of the indicated plasmids. At 24 h post-transfection, the medium was carefully removed, and the cells were washed with ice-cold PBS. Then the cells were lysed in 1 mL RIPA buffer at 4˚C on a horizontal shaker for 1 h. Cells lysates were centrifuged at 16000 × g at 4˚C for 20 min, then the supernatant was incubated with anti-Flag (M8823; Sigma-Aldrich, St. Louis,

MO, United States) or Myc magnetic beads (88842, Thermo Fisher Scientific, Waltham, MA, United States) overnight at 4°C. Finally, immunoprecipitates and total cell lysates (TCL) were analyzed by western blotting using the indicated antibodies.

## Ubiquitination inhibitor administration

HEK 293T cells were transfected with the intended plasmids. At 24 h post-transfection, the cells were treated with proteasome inhibitor MG132 (10 μg/mL) (S2619; Selleck, Shanghai, China) or dimethyl sulfoxide for 6 h. Total protein from the cells was extracted using RIPA buffer for western blot analysis.

## Ubiquitination assay

Transfected HEK 293T cells were washed twice with ice-cold PBS, lysed in buffer A (6 M guanidium-HCl, 0.1 M $Na_2HPO_4/NaH_2PO_4$ and 10 mM imidazole) and incubated with Ni2 +-NTA beads (Qiagen, Germantown, MD, United States) at 4°C on a horizontal shaker overnight. The beads were washed sequentially with wash buffer I (mix buffer A and wash buffer II to a ratio of 1 to 4) and wash buffer II (25mM Tris-Cl pH 8.0 and 20mM imidazole) three times. The bound proteins were eluted with wash buffer II and subjected to western blotting.

## 17β-estradiol administration

The *cyp17a1-/-;dmrt1-/-* fish were treated with 0.1 μg/L 17β-estradiol (E8875, Sigma-Aldrich) from 80 to 110 dpf. Fish ovaries were harvested for histological analysis. Ovaries dissected from the control fish and *cyp17a1-/-;dmrt1-/-* fish reared in the system water were used as positive and negative controls, respectively.

## Statistical analysis

Each experiment was performed in triplicate. Detailed information regarding the number of zebrafish used per experiment is provided for each experiment and corresponding figure. The results are expressed as the mean ± SD. All analyses were performed with the GraphPad Prism 6.0 software program and the differences were assessed using the Student's t-test for paired comparisons and one-way ANOVA, followed by Fisher's LSD test for multiple comparisons. For all statistical comparisons, a p value < 0.05 was used to indicate a statistically significant difference. Significant differences marked with asterisks and letters were analyzed using Student's t-test for paired comparisons, and one-way ANOVA followed by Fisher's LSD test for multiple comparisons, respectively.

## Supporting information

**S1 Fig. Targeted disruption of *cyp17a1* and *dmrt1*.** (A) Schematic representation of wildtype (*cyp17a1+/+*) and the mutant line of *cyp17a1* alleles in the first exon. (B) Schematic representation of the putative peptide of wildtype (*cyp17a1+/+*) and the mutated Cyp17a1 peptides. (C) Schematic representation of wildtype (*dmrt1+/+*) and the mutant line of *dmrt1* alleles in the sixth exon. (D) Schematic representation of the putative peptide of wildtype (*dmrt1+/+*) and the mutated Dmrt1 peptides.
(TIF)

**S2 Fig. *In situ* hybridization was performed on cryosections of presumptive ovaries using the sense probe of *fancl*.** (A) Control fish at 25 dpf. (B) *cyp17a1-/-;dmrt1-/-* fish at 25 dpf. Arrows point to the immature oocytes.
(TIF)

**S3 Fig. Additional target disruptions of *cyp17a1* and *fancl* in *dmrt1*-/- zebrafish.** (A) Schematic representation of the genomic locus for target disruption of *cyp17a1*. UTR, untranslated region. (B) The PCR results using the fish genomic DNA for genotyping *cyp17a1*, including *cyp17a1+/+*, *cyp17a1+/-* and *cyp17a1-/-*. (C) Schematic representation of genomic locus for target disruption of *fancl*. E, exon. (D) The PCR results using the fish genomic DNA for genotyping *fancl*, including *fancl+/+*, *fancl+/-* and *fancl-/-*.
(TIF)

**S4 Fig. The single domain mutation of Fancl did not affect its association with TP53.** (A) Myc-tagged Fancl with single domain mutation and Flag-tagged TP53 were transfected into HEK293T cells. Both the anti-Myc and anti-Flag antibody-conjugated agarose beads were used for immunoprecipitation. (B) Domain mapping revealed that the single domain mutation of Fancl did not affect its association with TP53. WDR, WD-repeat domain. ULD2, UBC-like domain 2. ULD3, UBC-like domain 3. CTD, C-terminal domain. IP, immunoprecipitation. IB, immunoblotting. TCL, total cell lysate.
(TIF)

**S1 Data. Source data for Fig 1M, 1N, 1O and 1V.**
(XLSX)

**S2 Data. Source data for Fig 2A, 2B, 2E, 2H and 2I.**
(XLSX)

**S3 Data. Source data for Fig 3M and 3C1.**
(XLSX)

**S4 Data. Source data for Fig 4B, and 4D.**
(XLSX)

**S5 Data. Source data for Fig 5B, and 5D.**
(XLSX)

**S6 Data. Source data for Fig 6G.**
(XLSX)

## Acknowledgments

We thank the China Zebrafish Resource Center for providing the zebrafish strain of *tp53* mutant line (IHB 136). We thank Mr. Wenyou Chen of the Institute of Hydrobiology, Chinese Academy of Sciences, for handling the zebrafish stock. We thank the Center for Instrumental Analysis and Metrology, Institute of Hydrobiology, Chinese Academy of Sciences, for technical assistance with section scanning and image capture.

## Author Contributions

**Formal analysis:** Yonglin Ruan, Xuehui Li, Xinyi Wang, Qiyong Lou, Xia Jin, Jiangyan He.

**Funding acquisition:** Gang Zhai, Zhan Yin.

**Investigation:** Yonglin Ruan, Xuehui Li, Xinyi Wang.

**Methodology:** Yonglin Ruan.

**Project administration:** Zhan Yin.

**Resources:** Jie Mei, Wuhan Xiao, Jianfang Gui.

**Software:** Xuehui Li.

**Supervision:** Gang Zhai, Zhan Yin.

**Validation:** Yonglin Ruan.

**Visualization:** Yonglin Ruan.

**Writing – original draft:** Yonglin Ruan, Gang Zhai, Zhan Yin.

**Writing – review & editing:** Gang Zhai, Zhan Yin.

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
