## [Decision Letter · Decision Letter 0]

24 Oct 2023

Dear Dr Zhai,

Thank you very much for submitting your Research Article entitled 'Insights into the all-testis differentiation without endogenous androgen signaling in zebrafish' to PLOS Genetics.

The manuscript was fully evaluated at the editorial level and by independent peer reviewers. The reviewers and editors appreciated the further deciphering of the sex determination pathway in zebrafish, evidence for the mechanism by which Fancl regulates Tp53, and the strength of the genetic analysis performed. Some concerns, however, were raised about the current manuscript that need to be addressed, including improving the quality of the in situs shown in Figure 1 and that quantitative analysis of expression levels of such in situs is not a rigorous method. There are many helpful comments from the reviewers to improve and clarify the figures and text throughout the manuscript. Please include also the Selman staging designations, as suggested, in addition to the current staging ones used. We do not require new experiments for the revision, except those needed to strengthen current results presented. Based on the reviews, we will not be able to accept this version of the manuscript, but we would be willing to review a much-revised version.

If you decide to revise the manuscript for further consideration at PLOS Genetics, please aim to resubmit within the next 60 days, unless it will take extra time to address the concerns of the reviewers, in which case we would appreciate an expected resubmission date by email to plosgenetics@plos.org.

We are sorry that we cannot be more positive about your manuscript at this stage. Please do not hesitate to contact us if you have any concerns or questions.

Yours sincerely,

Mary C. Mullins

Academic Editor

PLOS Genetics

Gregory Barsh

Editor-in-Chief

PLOS Genetics

Reviewer's Responses to Questions

**Comments to the Authors:**

Reviewer #1: The factors regulating sex determination and differentiation are not fully understood, even in organisms where dedicated sex chromosomes are present. In zebrafish, domesticated strains have lost the ZW based sex determination that is found in wild strains and as in other animals, the mechanisms of sex determination and sex-specific differentiation are not understood. Despite differences in upstream activators or triggers, differentiation of primary and secondary sex traits converges on regulation of androgen and estrogen levels. Remarkably, prior work in zebrafish established that normal spermatogenesis could occur in the absence androgen receptor or the Cytochrome P450, Cyp17a1a, when supplemented with progestin. In this work the authors investigated the relationship between a conserved male differentiation factor, Dmrt1, and Cyp17a1a, a key enzyme in androgen and estrogen synthesis. The authors took a genetic approach to examine the relationship between these factors and sex determination and differentiation and found that in contrast to loss of Cyp17a1a, which causes testis only development in zebrafish, loss of Dmrt1 in cyp17a1a mutants resulted in development of ovaries with only early-stage oocytes. They provide evidence that supplementing dmrt1;cyp17a1a mutants with estradiol supported oocyte development to later stages. Based on these observations the authors conclude that Dmrtt1 is required for testis differentiation and that estradiol is not required for ovary differentiation but is important for ovary development and maturation. Using an RNAseq approach, the authors determined that fancl expression was higher in dmrt1;cyp17a1a double mutants. Based on its elevated expression and because prior work in the field implicated Fanc family members in ovary development, and germ cell loss in ovary to testis transformation, the authors generated triple mutants lacking Dmrt1, Cyp17a1a, and Fancl. The authors found that these triple mutants, like cyp17a1a and fancl single mutants, developed exclusively as males, but unlike the fancl or cyp17a1a single mutants, the triple mutants had hypoplastic and abnormal testis development due to lack of Dmrt1. Tunel and germ cell analyses showed abnormal germ cell development and cell death in triple mutant testis. Since Tp53 was previously shown to suppress the loss of oocytes and sex reversal previously associated with mutations of Fancl family members, the authors investigated a potential role for Fancl in repressing Tp53. The authors provide evidence that Fancl and Tp53 bind to one another in 293 cells and identified the TMD domain as Tp53 as important for that interaction. Similar attempts to map the interaction domain of Fancl suggest that multiple domains may be involved and are sufficient for interaction with Tp53 in 293 cells. The authors performed in vitro assays in 293 cells in the presence and absence of Fancl and proteosome inhibitors and conclude that Fancl promotes degradation of Tp53 via the proteosome. Their model is that Fancl acts in the ovary to prevent oocyte loss by ubiquitinating Tp53 and promoting its degradation via the proteosome. Further they propose that Dmrt1 negatively regulates fancl levels to promote oocyte loss and later testis differentiation. Although there are a few missing methods descriptions and some missing statistical analyses, overall, the data in the manuscript are clear, with appropriate numbers of individuals examined and statistical analysis, and are mostly consistent with the authors conclusions. However, the exciting conclusions are somewhat inaccessible, especially to the broader scientific readership, in the current draft. In particular in the abstract, introduction, and discussion sections.

Major:

1. The major concern is with the writing, which would benefit from major revision. I have listed a few examples by section below; however, more comprehensive revision to set up the questions and exciting conclusions more clearly would greatly strengthen the manuscript and make it accessible to nonexperts.

2. Without measuring testosterone, estradiol in the various mutant contexts, it might be prudent to qualify some of the conclusions regarding “absence of androgen signaling” “absence of estradiol” etc. and instead stating that the specific genes are dispensable for ovary/testis determination, differentiation etc. This would allow for the possibility of other compensating enzymes or ways to make these steroid hormones. For example, it is fair to conclude that cyp17a1 is dispensable but is estradiol signaling completely eliminated or is it just reduced? Further, the authors and prior work indicate that some of the mutants have secondary sex trait deficits.

3. Regarding the conclusions regarding Fancl and Tp53, because gonads and germ cells were not examined, it should be clearly stated that these are interactions and activities in 293 cells. Further, the conclusions with respect gonad development should be qualified to state that they are extrapolated from 293 to gonad. For example, “ assuming similar regulatory relationships exist in the germline and gonads, these results provide novel insights into the regulatory network involved in Fancl functions for ovarian differentiation…”

4. Regarding the interaction domain of Fancl, do truncations that eliminate multiple domains abolish or attenuate binding?

5. For the western blots in Figure 3 and the supplemental figures, the abbreviation TCL should be defined in the legend.

6. Appropriate allele designations need to be provided for all mutants analyzed.

7. In Figure 1N and 2G, 4N “ratios” of various stages are reported. How were these ratios determined, from how many sections and individuals? Please clarify and describe this in the methods.

8. Figure 1R, relative to what? How was this normalized across samples? Please clarify and describe in the methods. Also, because comparisons are made between single and compound mutants, statistics should be included for Dmrt and double mutants as well.

9. Figure 1T-V, I understand that in situ on sections is challenging, but the image quality is not sufficient to reach a conclusion. In addition, quantitative conclusions are made based these experiments, but NBT:BCIP staining is not quantitative method since the researcher stops the reaction. Also, panel V is mislabeled as H on the figure.

10. Figure 5E, statistics should be included for dmrt1 single mutants and triple mutants to distinguish between an effect due to loss of Dmrt1 alone as opposed to synergistic effects.

11. The models in Figure 6 are helpful, but a few genotypes are combined under the same scenarios based on the labels, but they are not really equivalent. For example, in scenario F, ,Dmrt1 and AR would still be present in Fancl loss of function, but taking away Dmrt1 doesn't suppress. It would help also to distinguish between fertile and sterile ovaries and testis in the various models because these are not functionally equivalent outcomes.

12. In the methods under “transcriptome analysis” line 430, “At the indicated time points, total RNA from zebrafish gonads”. Please clarify if these were gonads or truncated bodies that contain gonads as described in the RNA extraction and qPCR section above.

13.

Minor:

14. Supporting figure S1 and S3 are nice but not necessary.

Writing:

Abstract:

1. “The laboratory strains of zebrafish lack a typical sex chromosome, and the gonadal differentiation genes downstream of a sex determination gene that regulates the animals develop as males or females, though conserved among vertebrates, is still unknown yet in zebrafish.” This is a complex sentence with many ideas that can be separated into simpler sentences.

2. From the abstract, “exclusive gonadal sex shift from the testes to the ovaries was caused by depletion of doublesex and mab-3 related transcription factor 1

(dmrt1) in cyp17a1-/- zebrafish.” This is potentially misleading as written as it might imply the double mutants make a testis that then becomes an ovary. And later “cyp17a1-/-;dmrt1-/- zebrafish, comparative gonadal transcriptome analysis revealed that the phenotypes shifting from testes to ovaries correlated with up-regulated gonadal fancl.”

3. “Mechanistically, the degradation of TP53 with the activation of its K48-linked polyubiquitination within its DNA-binding domain (DBD), mediated via the ubiquitin ligase Fancl, has been demonstrated.” This sentence is unclear.

4. “Taken together, our results revealed that Dmrt1 signaling determines testis fate for gonadal differentiation even in the absence of androgen synthesis in cyp17a1-/- fish, which antagonizes Fancl/TP53 signaling during the critical stage..” This is a complex sentence and seems inconsistent with the previous sentence states that the triple mutants male even in the absence of Dmrt1 and Cyp17a1.

Author summary:

1. “Without the presence of testosterone and estradiol in cyp17a1-deficient zebrafish, Dmrt1 signaling sufficiently promotes the all-testis differentiation via activating germ cell apoptosis mediated through TP53 by suppressing the expression of fancl.”

Introduction:

1. “Besides, unlike the female biased ratio phenotype and impaired spermatogenesis seen in ar-/- zebrafish [18], all-testis differentiation and normal spermatogenesis have been achieved in our cyp17a1-/-;ar-/- zebrafish.” This is unclear as written and seems inconsistent with the statements later in the paragraph thar ar and androgen is dispensable.

2. “Moreover, the interaction between Fancl and TP53 led to the discovery of the activation of K48- linked ubiquitination of TP53, which contributes to the survival of germ cells in zebrafish during gonadal differentiation.” This is misleading as written because the interactions and modifications were demonstrated for somatic cells, 293, and not for germ cells. Further it is unclear as written if the authors are referring to this work or other work.

3. “Dmrt1 can repress fancl expression, disrupt Fancl/TP53 interaction, promote apoptosis in germ cells during the critical gonadal differentiation window period, and determine all-testis differentiation, without the antagonistic effect of estradiol signaling in cyp17a1-/- zebrafish.” This sentence mixes multiple contexts and is not fully consistent with the data shown.

Discussion:

1. The authors report results including new results and refer back to the earlier figures 1 and 2. This is not typical of a discussion section which is typically a synthesis of results. Usually, the only figures referred to are new model or summary figures.

2. “However, when the nuclear progestin receptor (npgr) was deleted in cyp17a1-/-;ar-/- fish, the phenotype of all-testis differentiation was sustained even with impaired spermatogenesis in cyp17a1-/-;ar-/-;npgr-/- zebrafish.” “even with” should be “albeit with”

3. “This indicates that Dmrt1 signaling can still cancel the testis fate for gonadal differentiation, under testosterone and estradiol absence in cyp17a1-/- zebrafish.” This is inconsistent with the data shown and is potentially confusing phrasing. The data show Dmrt1 is required for testis fate not to cancel it.

4. “Therefore, it is reasonable to speculate that the synergistic effects on testis fate determination could be ensured by both discrete Dmrt1 and androgen signaling in our cyp17a1-/-;dmrt1-/- zebrafish”. This sentence is unclear as written.

5. “It is in lines with our previous view that Dmrt1 is required for the maintenance, self-renewal, and differentiation of male germ cells, and implying that in cyp17a1-/-;dmrt1-/- fish, the additional depletion of fancl resulted in sex reversal from ovaries to testes, which was compromised due to dmrt1-deficiency, which in turns, caused dysregulated male germ cells development afterwards.” This is a complex sentence with many ideas that can be separated into simpler sentences.

6. “Depletion of tp53 (IHB136, China Zebrafish Resource Center) did not lead to a female-biased zebrafish population and did not reverse all-testis differentiation in cyp17a1-/- zebrafish (S5 Fig), suggesting that TP53 signaling might only provide complementary, sustainable, but not decisive, roles for ovarian differentiation in zebrafish.” This is unclear. Do the authors mean to say that it is only a factor in the context of ovarian failure and not normal sex determination and differentiation.

7. “Our results further indicate that Dmrt1 is a key determining factor for testis differentiation in certain fish even in the absence of testosterone nor estradiol, partially via the suppression of fancl expression, which leads to elevated TP53-induced germ cell apoptosis in zebrafish (Fig 6).” This may be a bit confusing. It might help to add something about when Tp53-induced death of oocytes occurs during ovarian failure or in the bipotential gonad.

8. “S2 Fig. Every single domain mutation of Fancl did not affect its association with TP53.”

Reviewer #2: Previous work from this lab showed that, remarkably, cyp17a1 mutants, which cannot produce either the male-promoting hormone testosterone or 11KT, or the female-promoting hormone 17ß-estradiol, all develop as fertile males. Iin this present study they extend these findings by analyzing the phenotype of cyp17a1;dmrt1 double mutants. dmrt1 is a highly conserved regulator of male development and all dmrt1 zebrafish mutants are fertile females. The authors found that dmrt1;cyp17a1 double mutants develop as females. However, these females are presumably sterile as the oocytes they produce never progress past stage IB (primary growth stage). They further show that oocyte development can be rescued in the double mutants by estradiol treatment. By comparing the transcriptomes of normal and double mutant ovaries they identify many genes that are differentially regulation. One in particular is fancl, which was previously shown to be required for female development by suppression of Tp53-mediated germ cell apoptosis. Using proteomic analysis, they demonstrate that Fancl directly interacts with the DNA binding domain of Tp53, and that it can promote polyubiquitinate on K48. They further show that polyubiquitinate leads to the degradation of Tp53, thus providing a mechanistic model for the role of Fancl in ovary development. Finally, they show that removing fancl function in cyp17a1;dmrt1 mutants, which was not a trivial task given that fancl and cyp17a1 are on the same chromosome, lead to an all sterile male phenotype.

Two main conclusions can be drawn from this data: 1) In the absence of dmrt1 and androgen production, estrogen production is not required for ovary differentiation in zebrafish, but is required for oocytes to progress past stage IB. 2) Suppression of fancl expression by Dmrt1 is necessary for male/testis development. The data in the paper are clear and well presented, and the results will be of interest to a wide audience. The comments/suggestion below are mostly minor and intended to increase the clarity of the paper.

L47: …Dmrt1 signaling… (it is not a signal.) Better to say …Dmrt1 transcriptional regulation…

L68: “Cyp17A1 is a key enzyme involved in testosterone synthesis in animals.” It would be good to point out here that Cyp17a1 is required for both testosterone and estrogen production, as this is important to understanding the result presented here.

L102: For completeness in referencing primary publications, please add Dranow et al., 2016, which also analyzed cyp19a1a mutants.

L104: For completeness in referencing primary publications, please add Romano et al., 2020, which also analyzed the cyp19a1a;dmrt1 double mutants.

L101-111: A more appropriate reference for this statement is Tzung et al., 2015.

L 112: For completeness in referencing primary publications, please add Siegfried and Nusslein-Volhard, 2008, which also analyzed sex determination in dnd morphants.

L117: For completeness in referencing primary publications, please add Tzung et al., 2015.

L117: It would be more accurate to say “…specific levels of signals derived from oocytes…” as it

is oocytes, not pre-meiotic germ cells, that are required for female development (see also Rodriguez-Mari et al., 2010).

L132: More accurate to say: “…surviving oocytes…”

L141: “mysterious” is not a scientific term and should be avoided.

L165 & L280: For completeness in referencing primary publications, please add Webster et al., 2017, as they also showed dmrt1-/- fish are primarily female and the few males are sterile

Fig 1: This labels for the various oocyte stages appear accurate based on histological criteria, but the scale bar in Fig 1G does not appear to be accurate. Based on this bar, the FG oocytes are ~80-90 µm in diameter, which is the size of primary growth stage oocytes (Selman et al. 1993). FG oocytes should be 730-750 µm. Perhaps the bar is actually 50 µm. Please confirm all scale bars are accurate.

Fig 1: Please consider changing the oocyte stage designations (i.e. PG, EV, FG) to that of Selman et al., (1993) as it allows you to define the oocyte stages more precisely. For example, it is stated that cyp17a1;dmrt1 double mutant ovaries contain only pre-vitellogenic oocytes. However, pre-vitellogenic oocytes encompass pre-follicle (Stage 1A), follicle stage/primary growth stage (Stage IB), and cortical alveolus stage (Stage II) oocytes. Based on Fig 1L, cyp17a1;dmrt1 ovaries do not contain any oocytes that have progressed passed Stage IB. Thus, use of the Selman et al., staging allows a more precise description of the phenotype.

L170: “These results suggest…” This statement needs more context for the general audience to understand. Please explain the logic behind the conclusion that estradiol signaling is important for maturation but not differentiation.

Fig. 1T-H: How many independent gonads were examined and are the images in T-H representative? Please give n’s for each genotype. This is mainly a concern for the control gonads, because some may have already begun to transition to testes and may therefore have downregulated fancl. The image in U is of poor quality due to the many salt crystals that formed over the tissue. This panel should be replaced with a higher quality image. Please add labels for oocytes, if they are present (e.g. it looks like a stage IB oocyte is present in H).

L325: “Therefore it is…” The point being made by the sentence is not clear to this reviewer.

L334: cyp17a1-/- should be dmrt1-/-

Reviewer #3: The gene network that directs gonad to develop into ovary or testis is far from clear in zebrafish as well as in other vertebrates. In this study, the authors demonstrate that Dmrt1 determines the testis fate of gonads possibly via antagonizing Fancl/TP53 signaling even in the absence of androgen synthesis in cyp17a1-/- fish. Firstly, the authors analyzed the gonadal phenotypes of cyp17a1-/-, dmrt1-/- single mutants and cyp17a1-/-;dmrt1-/- double mutants in zebrafish. They showed that dmrt1 deletion reversed the all-testis phenotype of cyp17a1-/- zebrafish into all-ovary. Next, by comparing gene expression in the gonads of cyp17a1-/-;dmrt1-/- fish and control fish and rescue experiments by fancl (FA signaling gene) mutation, they showed that the up-regulation of fancl was responsible for the all-ovary development of cyp17a1-/-;dmrt1-/- zebrafish. Finally, they proved that fancl was involved in the regulation of germ cell apoptosis by mediating the degradation of the apoptotic factor TP53 via activating K48-linked polyubiquitination in the DNA binding domain. This study provides additional insights for zebrafish gonadal differentiation. The study is innovative, and the experiments are designed properly. However, several concerns should be addressed before publication.

Major:

1. The most attractive point of this manuscript is that the authors found Dmrt1 can antagonize Fancl/TP53 signaling to regulate gonad differentiation in cyp17a1-/- zebrafish with neither estrogen nor androgen. Considering that the expression of Fancl in developing germ cells and its function in TP53-mediated germ cell apoptosis and gonadal differentiation have been reported in zebrafish previously (Rodríguez-Marí et al., 2010), the authors need to focus their research on the relationship between Dmrt1 and Fancl, rather than studying how Fancl affects TP53. Is it possible to prove that Dmrt1 can directly suppress fancl expression and Fancl can mediate the degradation of Dmrt1 via polyubiquitination in vitro?

2. The transcriptome data displayed in the results section of this manuscript is not persuasive enough to convince readers that fancl up-regulation is the main reason for the all-ovary phenotype of cyp17a1-/-;dmrt1-/- zebrafish. Gonads should be sampled at early stage of gonadal differentiation rather than at 80 dpf, the up-regulation of fancl might be the consequence rather than the reason of sex reversal. Also, to find out the key gene that reversed the all-testis phenotype of cyp17a1-/- fish into all-ovary after dmrt1 deletion, gonadal gene expression should be compared between cyp17a1-/-;dmrt1-/- fish and cyp17a1-/- fish and between cyp17a1-/-;dmrt1-/- fish and dmrt1-/- fish, rather than between cyp17a1-/-;dmrt1-/- fish and wild type female fish.

3. It is widely accepted that estrogen but not androgens plays essential role in fish sex determination and differentiation. Blocking estrogen synthesis led to female to male sex reversal and estrogen administration led to male to female sex reversal in many fish species, including zebrafish. There has been no report showing that blocking androgen synthesis changes sex in fish. The all-testis phenotype of cyp17a1-/- and cyp17a1-/-;ar-/- zebrafish in this study should be explained by estrogen deficiency. It has nothing to do with androgen or progestin augmentation. The authors should change their description in the Title, Abstract (line 24-27) and Discussion (298-303) section of this manuscript to avoid misleading the readers.

4. Expression of fancl is upregulated in the dmrt1-/- fish. How about the gonadal development in the dmrt1-/-;fancl-/- fish? The authors should provide these data.

Minor:

1. Line 183-185, “The top enriched pathways were related to FA, steroid hormone biosynthesis, apoptosis, Notch signaling, Toll-like receptor signaling and mTOR signaling”. The order of the enriched pathway should be written according to either the gene numbers or the significance of pathways enriched. Also, the criteria for the identification of fancl as the key regulators need to be clarified.

2. Line 260-269, “However, the introduction ...... with a 37 bp deletion in the ninth exon”. This part should be move to the Materials and Methods section.

3. This MS focus on sex differentiation of zebrafish mutants. The authors should show the phenotype at different stages of gonadal development in these mutants. Not just only at adult stage.

4. In Figure 1Q, please show the expression of fancl in dmrt1-/- fish at 80 dpf? In Figure 1S, why fancl was not upregulated in the dmrt1-/- fish compared with control? In Figure 1T, no sense probe was shown as control. The signal was not specific in germ cell, but with a universal expression pattern. Why? It has been reported that fancl is specifically expressed in the germ cells (Rodríguez-Marí et al., 2010).

5. This study and other reports showed the dysgenesis testis in some dmrt1-/- fish. The germ cell was lost in these mutants, as shown in Figure 4J. Why Vasa-positive germ cell was observed in the dmrt1-/- fish in Figure 5B? In the gonadal histology of cyp17a1-/-;dmrt1-/-;fancl-/- fish, no cystic germ cells was observed as shown in Figure 4L. However, it was observed by Vasa staining in Figure 5D. Whether mutation of tp53 can rescue the germ cell survival in dmrt1-/- and cyp17a1-/-;dmrt1-/-;fancl-/- fish?

6. In Figure 3, the authors should demonstrate the protein interaction between TP53 and Fancl in vivo. Whether ubiquitination of TP53 showed difference in female and male gonads during zebrafish sex differentiation? Whether de-ubiquitination of TP53 can be detected in the fancl-/- fish? Fig 3D, the result of “IB: Myc” in the TCL group is missing.

7. In Figure 5, no obvious TUNEL/Vasa co-location was presented. The authors should show high quality figures. Whether apoptosis was observed in the cyp17a1 mutants?

8. Line 23-24, the sentence “Zebrafish has been used as a convenient model for teleost gonadal differentiation for many years” may not be necessary in abstract.

9. It should be noted that the sex reversal in mice are quite different from that in fish after cyp17a1 deletion. In mice, cyp17a1 deletion only resulted in underdeveloped accessory reproductive organs, leading to female-like external genitalia, the gonadal sex are not reversed. However, in fish, cyp17a1 deletion resulted in complete sex reversal. So, the results in mice cannot be used as evidence to support or discuss the role of androgen in gonadal differentiation or sex determination. The authors should consider to remove or adjust such description in Line 50, Line 68-70, Line 83, Line 381-383.

10. It is well known that double mutation of cyp19a1a and dmrt1 resulted in ovary development in zebrafish and other fish species. In fact, the cyp19a1a;dmrt1 double mutants were equivalent to the cyp17a1;dmrt1 double mutants in absence of estrogen synthesis. Both are males due to estrogen deficiency. Therefore, the analyses of cyp17a1;dmrt1 double mutants did not provide us more information about fish sex differentiation. However, the different female rate in dmrt1-/-, cyp19a1a-/-;dmrt1-/- and cyp17a1-/-;dmrt1-/- zebrafish deserves more investigation.

11. The introduction should be refined according to the findings of this research to make the questions clear. In Line 110, “in some teleosts” should be revised as “in zebrafish”, because the references cited in this paragraph only studied zebrafish.

12. Three main findings of this study are: Dmrt1 antagonizes Fancl/TP53 signaling to regulate gonad differentiation in zebrafish without both estrogen and androgen; Fancl regulates the degradation of TP53 via K48 mediated polyubiquitination; Different female rate was observed between dmrt1-/- and cyp17a1-/-;dmrt1-/- zebrafish. The discussion needs to be re-structured according to the main findings of this study to make it more logical.

13. The phenotype of cyp17a1-/-;dmrt1-/-;fancl-/- fish in this study was similar to that of the dmrt1-/-;rbmps2a-/-;rbmps2b-/- triple mutants. In zebrafish, maybe loss of genes related to germ cell development and survival results in the same phenotype. Please discuss it.

14. The scale bar in this manuscript should be adjusted to make it clear and consistent in each figure.

**Have all data underlying the figures and results presented in the manuscript been provided?**

Reviewer #1: Yes

Reviewer #2: Yes

Reviewer #3: None

PLOS authors have the option to publish the peer review history of their article (what does this mean?). If published, this will include your full peer review and any attached files.

Reviewer #1: No

Reviewer #2: No

Reviewer #3: No

---

## [Decision Letter · Decision Letter 1]

17 Jan 2024

Dear Dr Zhai,

Thank you very much for submitting your Research Article entitled 'New Insights into the All-testis Differentiation in Zebrafish with Compromised Endogenous Androgen and Estrogen Synthesis' to PLOS Genetics.

The manuscript was fully evaluated at the editorial level and by independent peer reviewers. The reviewers appreciated the changes made to the revised manuscript but identified numerous minor issues that we ask you address in a revised manuscript.

We therefore ask you to modify the manuscript according to the review recommendations. Your revisions should address the specific points made by each reviewer.

Yours sincerely,

Mary C. Mullins

Academic Editor

PLOS Genetics

Gregory Barsh

Editor-in-Chief

PLOS Genetics

Reviewer's Responses to Questions

**Comments to the Authors:**

Reviewer #1: This revised manuscript from Ruan and colleagues investigates the factors regulating sex determination and differentiation in zebrafish. In this work the authors investigated the relationship between a conserved male differentiation factor, Dmrt1, and Cyp17a1a, a key enzyme in androgen and estrogen synthesis.

In this revised version the authors have addressed the missing methods descriptions and statistical analyses. The authors have significantly revised the writing and added additional experiments to address concerns raised in the prior version; however, a few points, mostly related to writing, remain to be clarified.

1) Allele designations are needed throughout the manuscript.

2) Line 192: The point is lost in the sentence beginning “on the other hand..” Stage of oocytes is an important factor here. If Fancl is more highly expressed in early oocytes and DM gonads are blocked in oocyte progression, there will be an enrichment in early oocyte expressed genes in DMs compared to SM or controls, thus expression would remain high. If fancl is less abundant in older oocytes this will appear as increased expression in double mutants since they will lack the later stage oocytes and genes expressed at later stages will be less abundant or absent in DMs. If fancl has multiple roles, its expression may be dynamic in oocytes - is this seen in in situ or in the published single cell RNAseq datasets? Also, if Dmrt1 represses fancl in oocytes, their expression patterns might be reciprocal in oocytes e.g. when Dmrt1 is high, fancl would be low... was this relationship analyzed in the scRNAseq datasets?

3) Line 230: The sentence beginning “There results suggest..” The wording here is not entirely consistent with the data. These results suggest that Fancl is required for ovary differentiation. While fancl might be a target, negatively regulated by Dmrt1, the data suggest that other Dmrt1 targets are required for testis development since the DM testis is infertile.

4) Line 278-280: a reference is needed for the sentence on progestin signaling.

5) Line 309-310: “Indeed, we also observed a moderate up-regulation of fancl in dmrt1-/- fish at 17-23 dpf; however, it is not as significant as in cyp17a-/-;dmrt1-/- fish.” Can the possibility that this could be driven by differences in the stages of oocytes present in ovaries in these different mutant contexts be excluded? If not, please add a statement to qualify.

6) Line 342 and 343: Loss of Tp53 also did not rescue loss of Rbpms2 – Kaufman et al PLoSGenetics. Also, differentiation should be added along with cell survival and meiosis as failed differentiation will result in oocyte death.

7) In the “knockout lines” section of the methods, the guide RNA sequences and diagnostic primers should be provided. Alternatively, these sequences can be included in the supplemental table with the other primers.

Minor writing:

1) Line 47-48: “With the all-testis phenotype observed in cyp17a1-/- zebrafish, it has been suggested that androgen signaling is dispensable for testis differentiation in zebrafish.” Consider instead: “The all testis phenotype observed in cyp17a1-/- zebrafish, has led to the conclusion that androgen signaling is dispensable for testis differentiation in zebrafish”

1) Line 58: Instead of “entry into” consider “differentiation along”.

2) Line 59 and 60: “are complex due to its multiple types” Instead consider “are complex due to diversity within the species”

3) Line 66: Consider “influenced” rather than “enhanced”.

4) Line 77: “which” is not needed.

5) Line 79: “with” is not needed.

6) Lines 80-81: “Similarly these…” Consider instead “Similar phenotypes..”

7) Lines 93-95: The sentence beginning “It has been suggested that” should come later in the paragraph before the sentence on line 98 beginning “Dranow..”

8) Line 99: Need to be specific and state that this is Nanos3 because mutants in the different nanos genes have different phenotypes.

9) Line 112: “in mutants with 12 of the 17 FA genes…” Instead consider, “ for 12 of the 17 FA mutants.

10) Line 116: Instead of “continue” consider “persist or continue to increase”.

11) Line 118: A linker or transition phrase would be helpful here." One factor known to mediate apoptosis in the developing gonad is Tp53".

12) Line 119: Required seems more appropriate than responsible here since Tp53 alone is probably not the sole factor.”

13) Line 122: By “effectively” do you mean partially?

14) Line 127: By “with” do you mean "accompanied by", "associated with" or "along with"?

15) Line 154: “a signal” or “signals” rather than “the signal”.

16) Line 154: “induce” rather than “exist in inducing”.

17) Line 157: “regulate” rather than “to be required for”.

18) Line 157: “of” is needed after differentiation.

19) Line 160: should indicate that this is in wildtype.

20) Line 160: “Based on their results…” this sentence is a bit unclear. Consider something like “Among these candidate genes fancl was selected based on its early expression in gonads at 17 and 23 dpf and its abundant expression in presumptive female gonads”.

21) Line 163: consider “aligns” rather than “correlates”.

22) Line 168: “the up-regulated..” Consider instead “Upregulation of fancl genes was observed...”

23) Line 170: “Moreover” rather than “besides”.

24) Line 172: The sentence beginning, “To rule out the possibility..” There is a logic problem as written. It is not clear why a later change would be examined to control for a timing difference. The important issue is the ability to compare the stages present in the gonad to exclude the possibility that the expression difference is due to a difference in stages/types of cells present withing the gonads. Based on the images shown, it appears that there may be more early-stage cells present in the cyp17;dmrt double mutants compared to controls. To determine if increased expression is due to upregulation or more cells expressing, cells of the same stage should be compared.

25) Line 178 and elsewhere: Consider greater, higher, or more abundant rather than “upregulated” because expression could be higher due to increased expression (upregulation) or stability.

26) Line 180: “but exhibited no obvious signals of staining..” Consider instead, “and as expected no signals were detected”.

27) Line 181: “The” is not needed before comparative, and “in ovaries” should be “between ovaries”.

28) Line 182: “the” is not needed before “cyp17a1” and “fish of their” is not needed between female and control.

29) Line 184 “alterations of the expression levels..” Consider instead “expression level alterations of 424 genes”.

30) Line 186: “The majority of KEGG..” Consider instead “the most enriched KEGG pathways in..”

31) Line 220: The sentence beginning “However” is unclear. Do you mean “Unlike cyp;dmrt1 DMs which developed ovaries, cyp;dmrtq;fancl TMs developed testis with histologically apparent abnormalities including fb like somatic cells.....similar to those observed in...”

32) Line 224: The sentence beginning “The dissected testes..” can be simplified. “Accordingly, dissected testis of SM and TM were hypoplastic compared to controls.” Details about fixative should be in the methods section.

33) Line 228: “also has a hypoplastic testis…” Consider instead “testes were also hypoplastic and lack germ cells similar to dmrtt1-/-“.

34) Line 237: “which is an all-ovary differentiation mechanism..” Should be "context" rather than "mechanism".

35) Line 247: “mutant domain” should be “functional domain” unless the activity of the domain was altered.

36) Line 262: “albeit the” should be deleted.

37) Line 262 “the additional supplement..” consider instead “supplementation with estrogen..”

38) Line 265 “the” is not needed before treatment and “of” should be “with”

39) Line 266: “ovary” should be “ovaries”.

40) Line 267: “follicles of the” rather than “that of the”.

41) Line 268: “were” rather than “was” and “at PG..” rather than “in the PG”.

42) Line 268: “stages” rather than “stage”.

43) Line 275: “the” is not needed before cyp17a1

44) Line 277: “albeit” should be deleted.

45) Line 278: “The” is not needed before augmentation.

46) Line 278: Consider “proposed” rather than “found”.

47) Lines 280-282: This section is somewhat confusing as written. If supplementing with Progestin supports testis organization, then it makes sense that the receptor would be required for normal testis development. This result suggests progestin is important for normal organization of the testis and spermatogenesis, but not for determination of testis fate.

48) Line 286-287: By “an interactive function” do the authors mean antagonistic rather than interactive.

49) Line 289: “despite” rather than “albeit”.

50) Line 290: “the” is not needed before ovarian differentiation. In the same sentence, the data suggest this would be maintenance rather than ovary differentiation if ovaries are found among mutants prior to d60 but decline thereafter.

51) Line 293: “the ovarian appearance..” Consider instead “ovaries detected among”..

52) Line 297: “elevated” rather than “accumulated”.

53) Line 298: “incidence or frequency” rather than “rate”.

54) Line 300-301: “may exist in promoting gonadal embarkation on ovarian differentiation.” Consider instead “may promote ovary differentiation”.

55) Line 310: The sentence beginning “The highest up regulation of fancl” can be simplified “Elevated fancl..” Later in the same sentence starting on line 312 “, as their synergistic effect in inhibiting the relative” can be deleted. This sentence would be clearer if these thoughts were inverted. "The observed synergy between Dmrt1 and Androgen signaling in inhibiting fancl transcription in luciferase reporter assays suggests that elevated fancl in cyp17;dmrt1 DM may result from loss of these synergistic repressors.”

56) It would help to start the paragraph starting on line 315 with a broad sentence regarding cell death/oocyte survival and implicated pathways to introduce Tp53 which otherwise appears out of the blue.

57) Line 325 “it could not maintain ovary differentiation” can be simplified “due to impaired ovary differentiation”.

58) Line 326 “germ cell survival”. Do the authors mean oocyte? or are the authors proposing that Fancl might regulated GC survival in ovary and testis?

59) Line 345-346: “(in cyp17a1-/- ;tp53-/- fish, N=12).” This should be in the results section not the discussion.

60) Line 351: “maintains proper” rather than “properly maintains”.

61) Line 365: “elevated” rather than “accumulated”.

62) Line 369: “The” is not needed before determination.

63) Line 372: “the” is not needed before depletion.

64) Line 372: “for” is needed before “CYP17A1”.

65) Line 389: “respectively” is not needed.

66) Line 394 and 396: “used” rather than “adopted”.

Reviewer #2: The authors have addressed all of my previous concerns and I have no further concerns.

Reviewer #3: The authors have answered almost all my questions. The writing of the current manuscript becomes more readable and friendlier for readers. I think it would be better if the following suggestion can be considered before publication.

Abstract

1. Line 31-32: The description “The all-ovary differentiation phenotype observed in cyp17a1-/-;dmrt1-/- zebrafish can be rescued by additional depletion of fancl” feels like that the gonads of cyp17a1-/-;dmrt1-/-;fancl-/- fish will develop into all-ovary”.

2. Line 26-32: In the present study, the female-biased sex ratio phenotypes were positively correlated with upregulated levels of Fanconi anemia complementation group L (fancl) in the gonads of doublesex and mab-3 related transcription factor 1 (dmrt1)-/- and cyp17a1-/-;dmrt1-/- fish, as fancl transcription being synergistically inhibited by Dmrt1 and androgen signaling. The all-ovary differentiation phenotype observed in cyp17a1-/-;dmrt1-/- zebrafish can be rescued by additional depletion of fancl”. Is it possible to write in the following way: “In the present study, the female-biased sex ratio phenotypes......, and additional depletion of fancl in cyp17a1-/-;dmrt1-/- zebrafish reversed the female-biased sex ratio to...... Luciferase assay revealed synergistic inhibitory effect of Dmrt1 and androgen signaling to fancl transcription”. “as fancl transcription being synergistically inhibited by Dmrt1 and androgen signaling” should be “as fancl transcription is/was synergistically inhibited by Dmrt1 and androgen signaling”

3. Line 33: Change “Tumor Protein p53” into “Tumor protein p53”.

Author summary

4. Line 53-55: “Our current study provides new insights into the interactive signals that regulate sexual fate determination with impaired androgen and estrogen synthesis in teleosts”. It would be better to remove the description “with impaired androgen and estrogen synthesis”.

Introduction

5. Line 67: Change “including” to “especially”.

Results

6. Line 205-213: “cyp17a1-/-;dmrt1-/- fish were derived...... then inbred to generate triple homozygotes (cyp17a1-/-;dmrt1-/-;fancl-/-)”. I think this part can be moved to Materials and methods section.

7. Line 231-232: “strengthen the notion that fancl is the downstream regulator of dmrt1 in determining testis fate”. Maybe it is not suitable to say that fancl is the downstream regulator of dmrt1 in testis fate determination. Is fancl expressed in testis? Is fancl downstream of dmrt1 in testis? Maybe it can be written as “highlighting the antagonistic role of fancl and dmrt1 in determining gonad sex”.

Figures

8. Figure 2C, D: Please quantify the increased expression observed for fancl.

Figure legends

9. Line 840: “90 dpf” should be “110 dpf”.

**Have all data underlying the figures and results presented in the manuscript been provided?**

Reviewer #1: Yes

Reviewer #2: Yes

Reviewer #3: None

PLOS authors have the option to publish the peer review history of their article (what does this mean?). If published, this will include your full peer review and any attached files.

Reviewer #1: No

Reviewer #2: No

Reviewer #3: **Yes: **Deshou Wang

---

## [Editor Report · Decision Letter 2]

5 Feb 2024

Dear Dr Zhai,

We are pleased to inform you that your manuscript entitled "New Insights into the All-testis Differentiation in Zebrafish with Compromised Endogenous Androgen and Estrogen Synthesis" has been editorially accepted for publication in PLOS Genetics. Congratulations!

Yours sincerely,

Mary C. Mullins

Academic Editor

PLOS Genetics

Gregory Barsh

Editor-in-Chief

PLOS Genetics

Comments from the reviewers (if applicable):

**Data Deposition**

http://datadryad.org/submit?journalID=pgenetics&manu=PGENETICS-D-23-01005R2

**Press Queries**

---

## [Editor Report · Acceptance letter]

22 Feb 2024

PGENETICS-D-23-01005R2 

New Insights into the All-testis Differentiation in Zebrafish with Compromised Endogenous Androgen and Estrogen Synthesis 

Dear Dr Zhai, 

We are pleased to inform you that your manuscript entitled "New Insights into the All-testis Differentiation in Zebrafish with Compromised Endogenous Androgen and Estrogen Synthesis" has been formally accepted for publication in PLOS Genetics! Your manuscript is now with our production department and you will be notified of the publication date in due course.

With kind regards,

Judit Kozma

PLOS Genetics

On behalf of:
